# Three-and-a-half million years of Tibetan Plateau vegetation dynamics in response to climate change

Yan Zhao [1,2] ✉, Feng Qin[1], Qiaoyu Cui[1], Quan Li[1], Yifan Cui [1], H. John B. Birks[3,4], Chen Liang[1,5], Wenwei Zhao[6], Huan Li [6], Weihe Ren[1,7], Chenglong Deng [2,8], Junyi Ge[9], Yanfen Kong[1], Yaoliang Liu[1,10], Zhiyong Zhang[1,11], Jiawu Zhang[12], Maotang Cai[1,13], Haicheng Wei[14], Hongyi Qiu[1], Haitao Xu[1], Hanfei Yang[1,15], Chunzhu Chen [6], Shilong Piao [16] & Zhengtang Guo[2,8]

The Tibetan Plateau supports the largest alpine meadow ecosystem globally. It is considered extremely vulnerable to global warming. Knowledge of past vegetation dynamics under similarly warm climates could shed insights into where the tipping point for regime shifts may lie. We report a continuous multicentennial-resolved pollen record for the last 3.5 Myr from a lake sediment core retrieved from the Zoige Basin (~3,350–3,450 m above sea level) on the eastern Tibetan Plateau. It reveals a detailed picture of the vegetation dynamics across several timescales using the approaches of biomization, numerical analysis, statistical modelling and vegetation simulations. These lines of evidence show that vegetation underwent transformation from stable forest in the mid-late Pliocene Period (3.5–2.73 million years ago (Ma)) to codominance of forest and steppe in the early Quaternary Period (2.73–1.54 Ma) and to a meadow-dominated ecosystem after ~1.54 Ma, along with glacial–interglacial and millennial-scale grassland–forest shifts. These vegetational changes were largely controlled by temperature change. A global warming of ~2–3 °C is the most important threshold for the forest expansion and meadow resilience loss on the Tibetan Plateau. By analogy to the past, we suggest that, without major reductions in greenhouse gas emissions, the current Tibetan Plateau meadow is at risk of major transformation.

The Tibetan Plateau (TP) supports a large and globally unique forest–meadow–steppe ecosystem (Extended Data Fig. 1a), which provides several ecosystem functions and services[1,2]. However, how this ecosystem would respond to future global warming remains unclear, largely due to the limited length of observational data and the uncertainty of vegetation models[3]. Long-term palaeovegetation records spanning different climate and $CO_2$ scenarios would be of particular value in predicting future vegetation dynamics and gauging model sensitivity[4,5].

Pollen assemblages are widely used to reconstruct past vegetation[6]. To date, except for the pollen record of ~1.7 million years ago (Ma) from the eastern TP previously reported[7], most other high-resolution pollen data are restricted to the latest Pleistocene Period[8], while data resolution and quality for older time periods[9,10] are insufficient to resolve continuous and detailed vegetation changes across time. The early Pleistocene and Pliocene Periods are considered among the best analogues for evaluating the effect of future global warming[4,11] as they are characterized by warmer climate, a smaller Greenland ice sheet and

**Fig. 1 | Location and regional settings of the Zoige Basin, eastern TP. a**, Location of the Zoige Basin. The yellow line delineates the extent of TP and the base map is derived from Esri World Imagery (https://www.arcgis.com/home/item.html?id=10df2279f9684e4a9f6a7f08febac2a9). **b**, Bathymetric map of Zoige region, with location of drilling site (marked by red dot) and digital elevation (m a.s.l.) (http://www.geodata.cn/). **c**, Modern vegetation in the Zoige Basin and the surrounding mountains based on data from http://www.geodata.cn/, modified from ref. 7. **d**, Elevational belts of vegetation types in the basin and surrounding mountains, suggesting that temperature controls tree distribution[7,12]. The elevation gradient is not equally scaled to provide a clearer illustration of the high-elevation vegetation belts.

higher $CO_2$ concentration. The mid-Pliocene interval is of particular value as it is marked by a global temperature of ~2–3 °C warmer relative to pre-industrial (PI) values and a $CO_2$ level of ~380–450 ppmv, very similar to future global scenarios[4,11]. High-resolution pollen time series from the TP since this critical period are lacking.

We present a continuous pollen record for the last 3.5 Myr with multicentennial resolution derived from lacustrine sediments in the Zoige Basin on the eastern TP (Figs. 1 and 2; Methods). It represents the longest continuous pollen record with similar resolution at the global scale. The results provide a detailed picture of vegetation composition and resilience and how they responded to the wide ranges of climate and $CO_2$ forcings in the past.

## Results

### Regional setting, drilling and core analyses

The Zoige Basin on the eastern TP is a tectonic basin (Fig. 1b) occupied by a huge lake until it drained ~28 thousand years ago (ka BP) (refs. 7,12).

Mean annual temperature (MAT) at the nearby meteorological station (~3,440 m above sea level (a.s.l)) is -1.4 °C, with a mean July temperature of ~11 °C and a mean January temperature of ~−9 °C. Mean annual precipitation is ~650 mm with most precipitation falling as rain during the summer months owing to the influence of the Asian monsoon[7].

Currently, local vegetation is subalpine meadow dominated by *Kobresia* and *Carex* spp. as well as grass, forb and shrub taxa, while the surrounding mountains are within the subalpine dark coniferous forest belt (situated at ~3,000–3,800 m a.s.l) (Fig. 1c,d), which is dominated by *Picea* and *Abies*, with broadleaf trees such as *Betula* and *Quercus*[12]. This forest belt is bounded by alpine shrubland and meadow and <~3,000 m by montane conifer and broadleaf mixed forest. An elevational vegetation gradient is clear in this region (Fig. 1)[7,12,13], as also shown in our modern pollen data[14]. The modern vegetation distribution in this alpine forest–meadow ecotonal region is primarily controlled by temperature, given the relatively high precipitation and low evaporation[7,12,15].

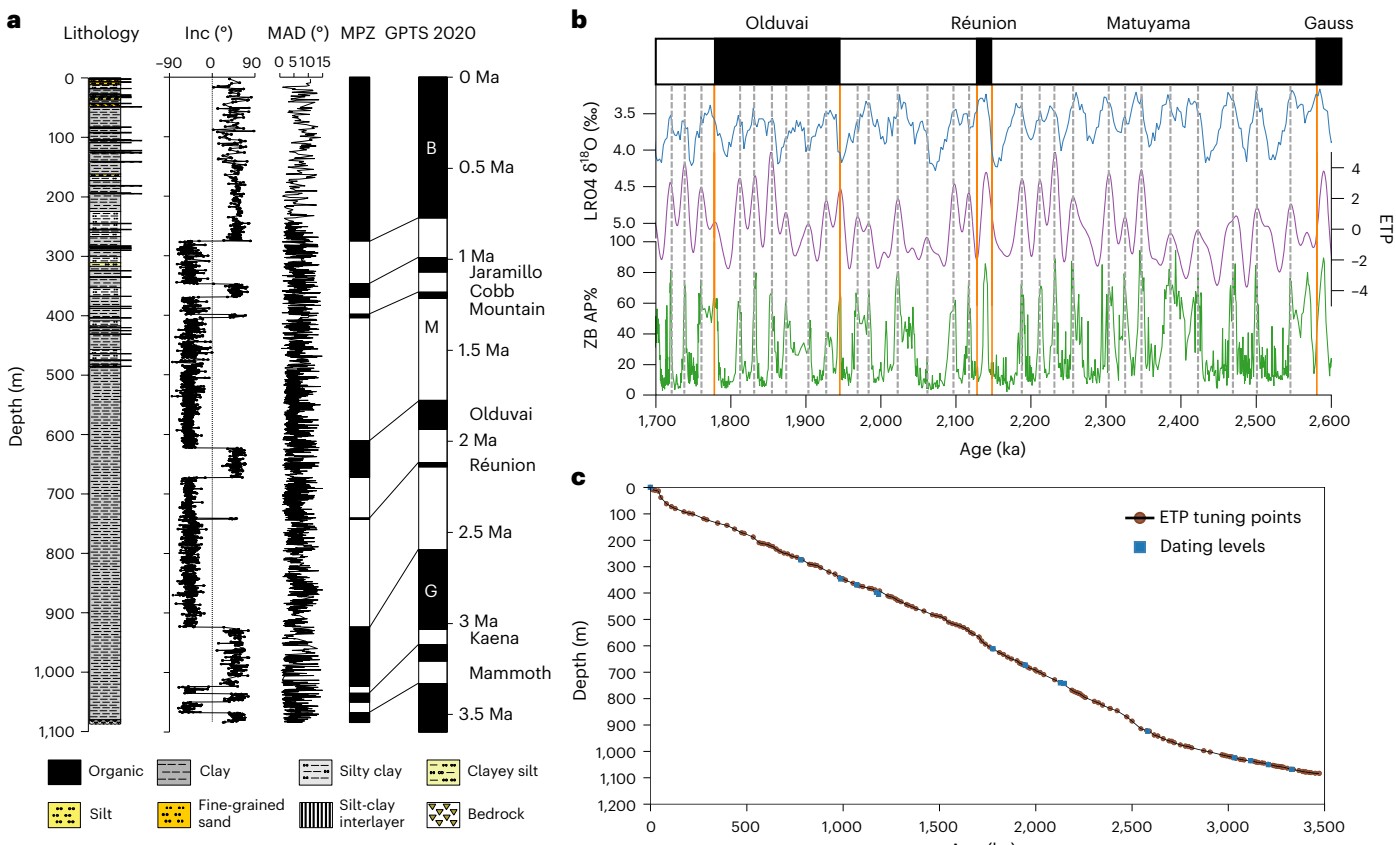

**Fig. 2 | Lithostratigraphy, magnetostratigraphy and chronostratigraphic plot of the core ZB19-C1. a**, Lithostratigraphy and magnetostratigraphy. The panels from left to right: lithology, inclination (Inc), maximum angular deviation (MAD), magnetic polarity zonation (MPZ) and correlation with the geomagnetic polarity timescale (GPTS 2020)[46]. Clear geomagnetic reversals mark the Brunhes/Matuyama and Matuyama/Gauss boundaries, as well as the Jaramillo, Cobb Mountain, Olduvai and Réunion subchrons in the Matuyama chron and the intra-Gauss reversals of the Kaena and Mammoth subchrons. **b**, Chronostratigraphic plot for the time window from 1,700 to 2,600 ka illustrating the definition of tie points used for creating the age model. Panels from top to bottom: LR04 δ[18]O stack[17], ETP generated by normalizing and averaging variations in E, T and P[16] and AP%. Solid orange and grey dashed lines denote the geomagnetic reversals and tie points, respectively. **c**, Correlation between the initial age model and the ETP tuning age. The initial age model using linear magnetostratigraphy controls yields similar sediment accumulation rates to those of the ETP chronology.

A 1,084.67-m core ZB19-C1 (33° 58′ 03″ N, 102° 20′ 09″ E, 3,442 m a.s.l) reaching to the basal rock was obtained in the sedimentation centre of the basin. It mainly consists of fine-grained freshwater lacustrine sediments, except for two episodic fluvial sandy layers at the top (Fig. 2a). Independent age controls derived from magnetostratigraphy provide an initial chronological framework, according to which the core extends back to ~3.5 Ma (Methods; Fig. 2 and Supplementary Table 1). A more detailed age model was constructed by tuning the arboreal pollen abundances (AP%) to an ETP record which is generated by normalizing and averaging variations in eccentricity (E), tilt (T) and reversed precession (P)[16] (Fig. 2b,c). The top 583 m of the core shows identical lithology to the previously drilled core ZB13-C2 (~400 m away from ZB19-C1) covering the last ~1.72 Myr (ref. 7). The AP% in the overlapping interval of ~1.72–1.58 Ma of the two cores is identical. These two nearby cores are thus combined to generate a uniquely continuous palynological archive (totally 5,000 samples with an average ~700-yr time resolution) covering the past 3.5 Myr.

### Vegetation transformations and shifts across timescales

The pollen data for the last 3.5 Myr of the Zoige Basin document three major vegetation transformations at ~2.73, ~1.54 and ~0.62 Ma (Methods) superimposed on a gradual long-term trend from forest to grassland (Fig. 3 and Extended Data Figs. 2a,b and 3a). Before the first transformation at ~2.73 Ma, AP% ranges from 60% to >90% with only a few samples having values of <50%. *Picea*, *Abies*, *Pinus* and *Tsuga*

dominate the tree taxa with the occasional occurrence of *Keteleeria* which currently grows at lower elevations. The tree taxon richness estimated by rarefaction analysis (Methods) shows highest values in the last 3.5 Myr with a median of 7 (Fig. 4a and Extended Data Fig. 4b). After 2.73 Ma, steppic herb (Poaceae, *Artemisia* and Amaranthaceae) and shrub associations (mainly *Hippophae* and Rosaceae) were established during glacial times at the cost of trees. *Tsuga*, which prefers warmth, decreases sharply from ~2.5 Ma. Tree taxon richness decreases to mostly <7 (median 6).

The second transformation at ~1.54 Ma is marked by a sharp increase in the abundance of typical meadow components; for example, Poaceae, Ranunculaceae, *Polygonum*, *Thalictrum* and various Asteraceae taxa, along with Cyperaceae (Fig. 3). Tree taxon richness further decreases to values mostly <6, while herb taxon richness clearly increases (Fig. 4a and Extended Data Fig. 4b).

The third major transformation occurring at ~0.62 Ma is characterized by a sharp AP% decline[7]. It coincides with marine isotope stage (MIS) 16, one of the largest Pleistocene glaciations[16,17]. Low tree taxon richness is coeval with the lowest tree abundances, particularly for the interval after the Mid-Brunhes Event at ~0.43 Ma (ref. 18).

Biomization analysis quantitatively confirms the transformations of the dominant megabiomes (Methods) over the last 3.5 Myr (Fig. 3b). Dominant megabiomes are forest, meadow, steppe and shrubland, while desert-steppe is only identified in a few samples. Forest vegetation occurred in ~91% of the Pliocene, suggesting a generally stable

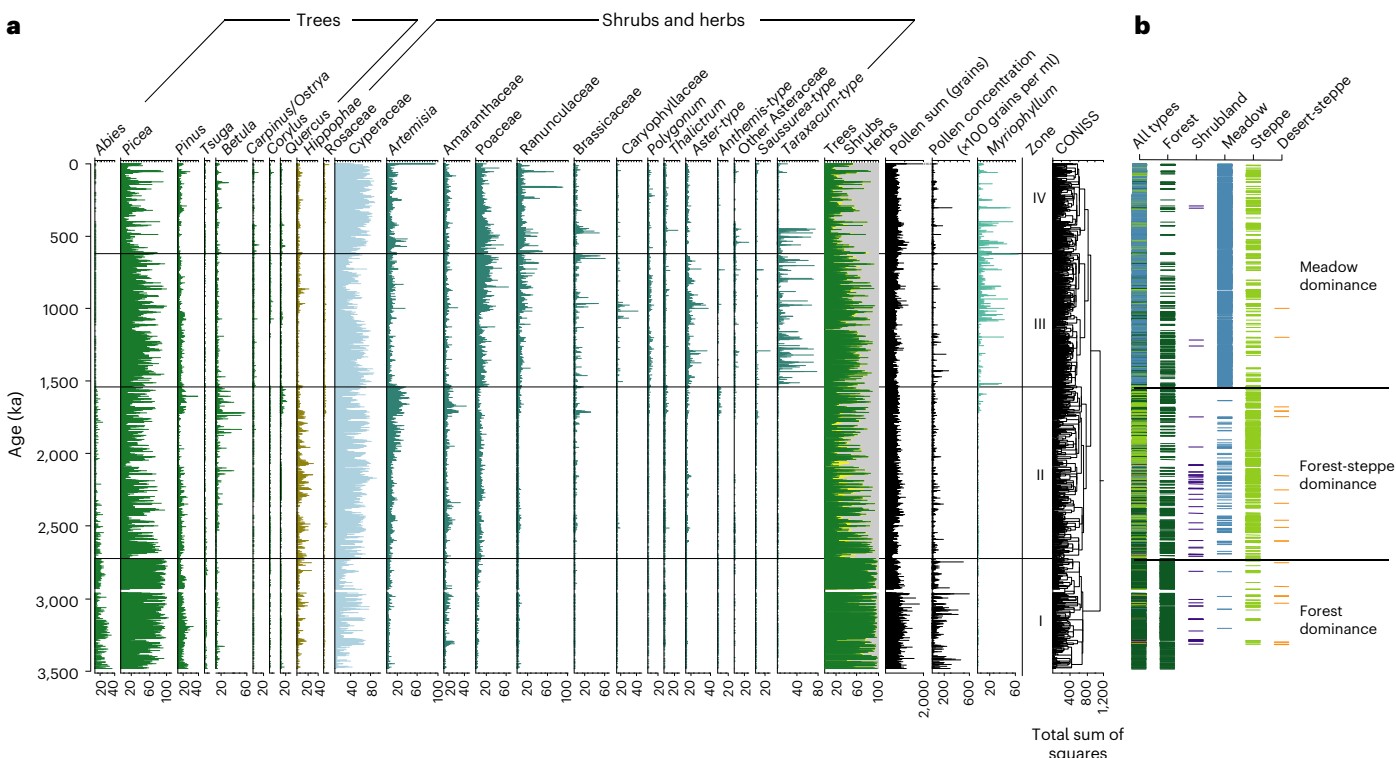

**Fig. 3 | Pollen percentage diagram of major taxa and biome reconstruction from the combined core of ZB13-C2 and ZB19-C1 from the Zoige Basin. a**, Pollen percentage. Tree taxa mainly include *Abies*, *Picea*, *Pinus*, *Betula* and deciduous *Quercus*. Major meadow/steppe taxa consist of Cyperaceae, *Artemisia*, Poaceae, Ranunculaceae and various taxa from Asteraceae. The pollen zonation is based on CONISS results aided by a multivariate regression tree analysis, which shows major changes at 2.73, 1.54 and 0.62 Ma. **b**, Combined megabiomes. The results indicate the establishment of typical meadow starting from ~1.54 Ma.

regime, but this proportion progressively declined to ~36%, ~18% and ~5% in the intervals of 2.73–1.54, 1.54–0.62 and 0.62–0 Ma, respectively (Extended Data Fig. 2b). Shrubland and steppe were codominant with forest at 2.73–1.54 Ma, occurring for ~55% of the time. Meadow occurred on the Plateau with only <1% during 3.5–2.73 Ma, for 9% from 2.73–1.54 Ma, before sharply increasing to >70% after 1.54 Ma. These suggest that the dominance of meadow on the eastern TP was not established until ~1.54 Ma.

Superimposed on these long-term changes are vegetation changes on orbital scales, with forest generally dominating in interglacials and meadow/steppe/shrubland dominating in glacials. However, in the Pliocene, forest is the predominant biome even in the glacial stages (except in a few deep glacials, for example, M2 and KM2) (Extended Data Fig. 5), while in the interval of 1.54–0 Ma, meadow occurs frequently even during interglacials. Wavelet analysis on AP% reveals persistently strong ~20-kyr periodicities, but with increased ~40-kyr and ~100-kyr glacial–interglacial signals in the intervals of 2.73–0.62 and 0.62–0 Ma, respectively (Fig. 4b).

On the millennial timescale, frequent vegetation fluctuations are reflected by AP% variabilities (Fig. 4b). These signals are weak for the Pliocene, but enhanced in the glacials of 2.73–1.54 Ma. Since ~1.54 Ma, remarkable millennial fluctuations of AP% occur in both glacials and interglacials, suggesting high instability of the forest ecosystem. In contrast, meadow became more stable after ~1.54 Ma, although with frequent fluctuations in the abundances of different herbaceous taxa.

## Discussion
### Global drivers of eastern TP vegetation changes
The observed long-term vegetation transformations from biomes dominated by forest, to forest–steppe, to meadow are unequivocally consistent with the major global cooling in relation to Northern Hemisphere glaciation as well as $CO_2$ decrease at ~2.73, 1.6–1.5 and 0.9–0.6 Ma

(refs. 17,19,20), suggesting a causal link (Fig. 4a). The larger variability amplitude and changing ice volume around these transitions further support this mechanism (Extended Data Fig. 6), particularly, the global climate stepwise descent into ice ages over time, as revealed by the marine $\delta^{18}O$ (ref. 17) and global mean surface temperature (GMST) records[20] (Extended Data Fig. 6b,c).

The vegetation transformations around these climate transitions are also observed in other terrestrial pollen records across latitudes (Extended Data Fig. 7). A striking forest decline at ~2.7 Ma is revealed by Lake El'gygytgyn pollen data (~490 m a.s.l) spanning ~3.6–2.2 Ma from the Arctic region[21,22]. Pollen data from the Bogotá basin in the tropical high Andes (~2,550 m a.s.l) spanning the last 2.25 Myr shows large decreases of AP% at ~1.5 and ~0.9–0.6 Ma, although the decline around 1.5 Ma is also associated with the sedimentary setting change due to rapid subsidence[23]. A sharp forest decline starting at MIS 22–16 is also recorded by a ~1.4-Myr pollen record from Lake Ohrid (~690 m a.s.l)[24] and Tenaghi Philippon (~40 m a.s.l)[25] in mid-latitude Europe. Nonetheless, these results indicate that the cooling around ~2.7, ~1.5 and ~0.9–0.6 Ma may have caused pervasive vegetation transformations for both mountain and low-elevation regions at a global scale, although more continuous and high-resolution pollen time series covering these time spans are greatly needed.

The strong ~20-kyr cyclicity in AP% before 2.73 Ma points to the predominant control of low-latitude summer insolation[16], which could impact both local heating and transferred heating and moisture via monsoon circulation. However, the modulation of high-latitude ice boundaries becomes reinforced across the Quaternary, causing stronger ~40- and ~100-kyr cycles in AP%. The fundamental ~20-kyr cyclicities in AP% are different from the four above-mentioned pollen records, which show much stronger glacial–interglacial cycles, highlighting the particular ~20-kyr vegetation feature of monsoonal regions.

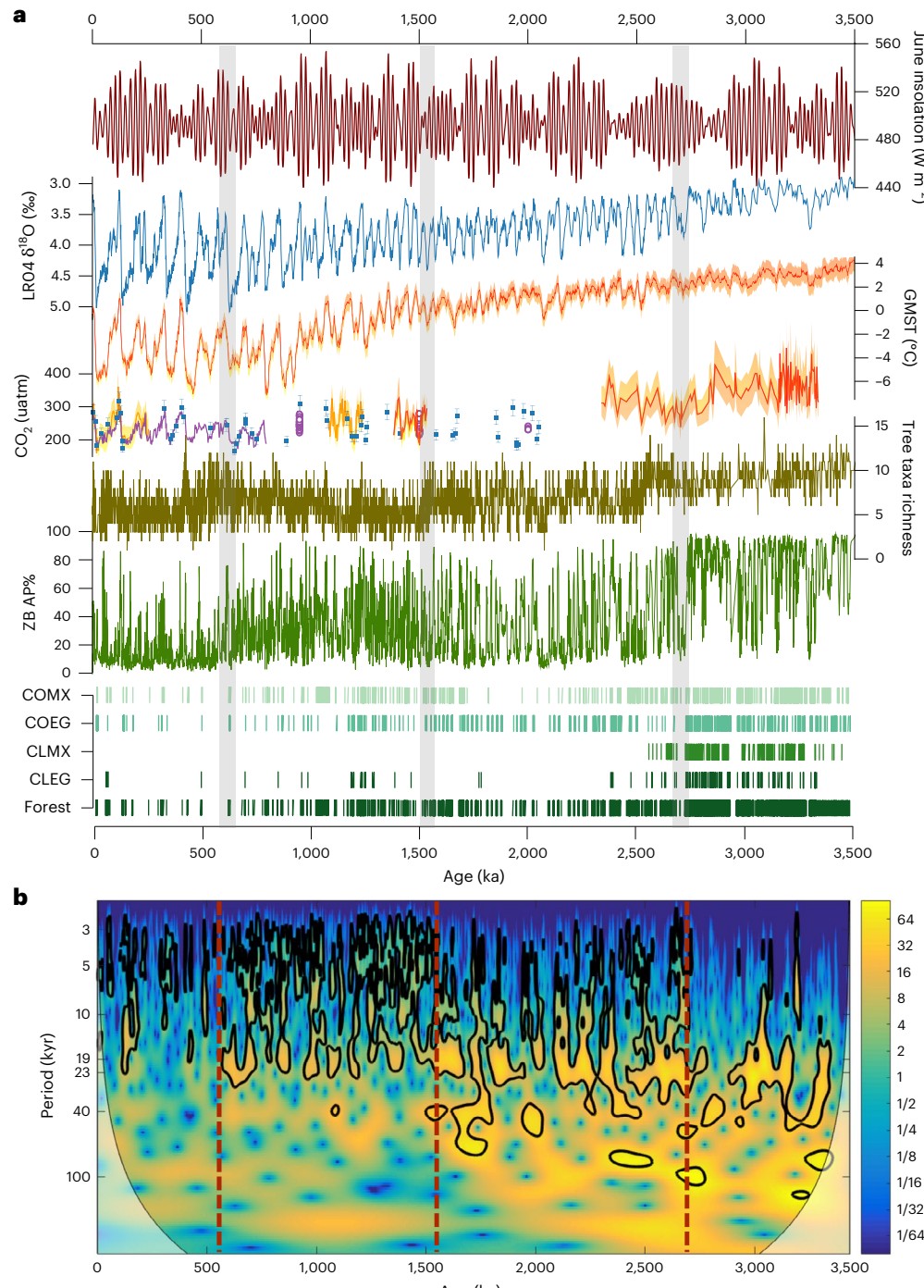

**Fig. 4 | Vegetation changes of the eastern TP on various timescales and correlation with global climate. a**, Correlation of vegetation records with insolation, ice volume, global surface temperature and $CO_2$ records. From top to bottom: mean June insolation at 30° N (ref. [16]); LR04 benthic $\delta^{18}O$ stack[17]; GMST relative to PI values with 1$\sigma$ uncertainty[20]; $CO_2$ records including $\delta^{11}B$-based data from the Caribbean[103] and eastern tropical Atlantic[104] (orange lines, with 2$\sigma$ error envelopes), low-resolution $\delta^{11}B$ record from the equatorial Atlantic[105] (blue squares, with 2$\sigma$ error bars), ice core $CO_2$ measurements from blue ice[106] (purple circle) and ice core compilation[107] (purple line); and rarefied tree taxa richness, AP%, as well as forest biomes from the Zoige Basin. **b**, Continuous wavelet transform results of AP% data. The data were resampled at equally spaced 1-kyr intervals and detrended before analysis. AP% shows a persistently strong ~20-kyr cycle, along with progressively strengthened cyclicities of ~40 and 100 kyr. Millennial-scale variabilities become strong after 1.54 Ma. The colour scale indicates the wavelet power, with hotter colours representing stronger power. CLEG, cold evergreen needle-leaved forest; CLMX, cold-temperate evergreen needle-leaved and mixed forest; COEG, cool evergreen needle-leaved forest; and COMX, cool mixed forest.

On a shorter timescale, the highly unstable forest ecosystem regime since ~1.54 Ma (Fig. 4) can be attributed to the remarkable strengthening of millennial climate variabilities as indicated by the North Atlantic ice-rafted debris that is also associated with ice boundaries[26]. Interglacial climate instability has also been revealed by marine and terrestrial records in the later part of the Quaternary[27,28], which can explain the biome fluctuations on the eastern TP in both glacials and interglacials since 1.54 Ma. Our pollen data with an average resolution of ~700 yr therefore provides a unique record for understanding the sensitive response of TP vegetation to global millennial climate variabilities.

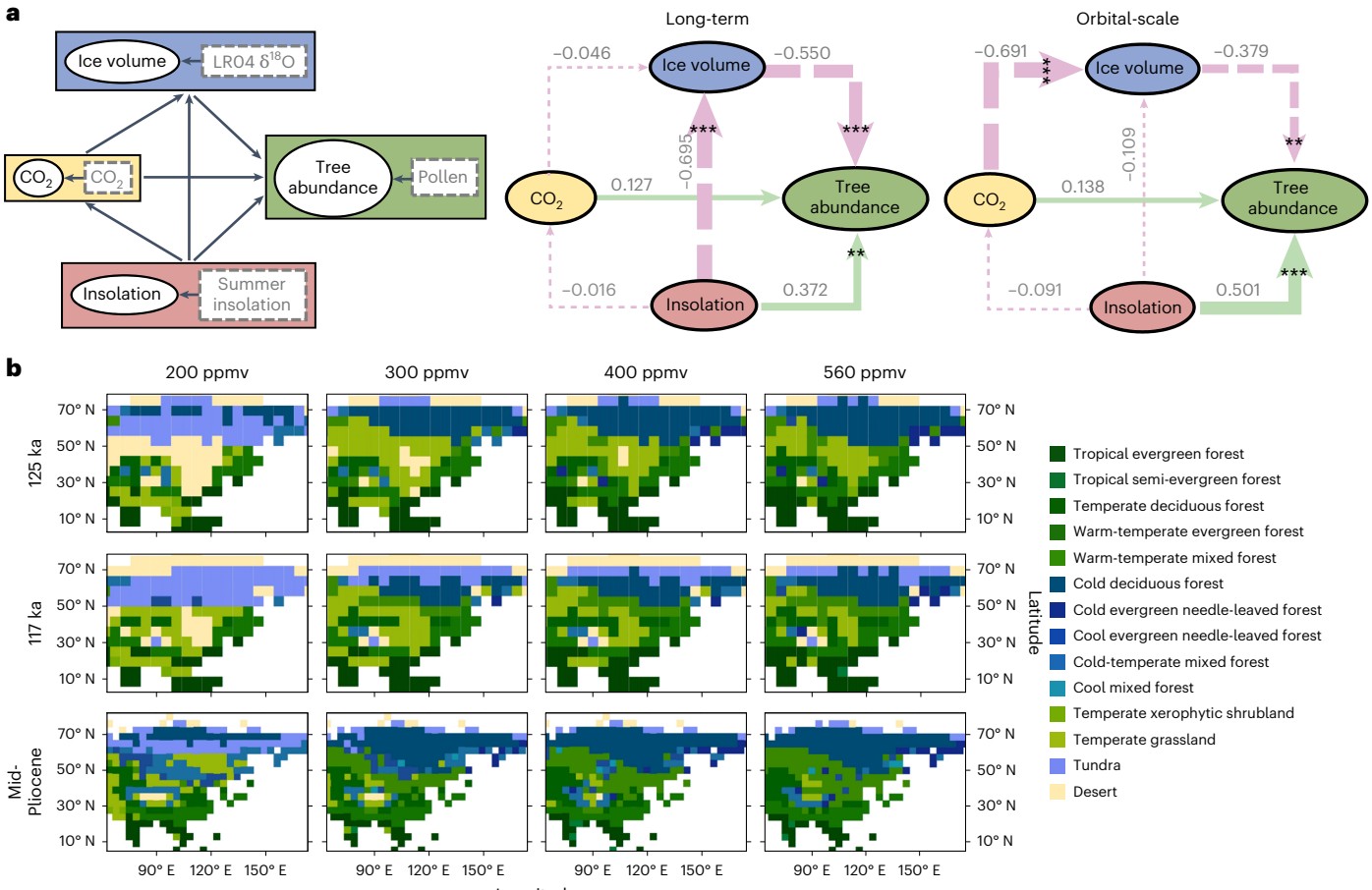

**Fig. 5 | Global drivers of vegetation dynamics on various timescales in the eastern TP. a**, Assessment of past relationships between TP vegetation, summer insolation, global ice volume and atmospheric $CO_2$ using an SEM approach. Left, hypothesized relationships examined in this study, where measured variables are indicated by square grey boxes; middle, long-term change based on all AP% data; right, orbital-scale changes based on bandpass (13–250 kyr) filtered AP%. Arrows in the diagram represent direct causal influences of one variable on another. Arrow colour/form indicates positive (green, solid) and negative (pink, dashed) relationships and arrow thickness represents the absolute strength of the relationships. Numbers denote the *r* values. Significance testing: **0.001 < *P* < 0.01 and ***P* < 0.001. **b**, Coupled climate–vegetation

model simulations for different orbital and $CO_2$ configurations of the mid-Pliocene warmth and the last interglacial–glacial. The three panels from top to bottom are a warm boreal summer orbit at 125 ka, an orbital configuration at 117 ka producing relatively cold boreal summers and a warm summer orbit in the mid-Pliocene. The four panels, arranged from left to right, display the results corresponding to $CO_2$ concentrations of 200, 300, 400 and 560 ppmv, respectively. COMX, COEG, CLMX and CLEG are the major forest types. The simulations place the pollen-based vegetation reconstruction into a spatial and temporal context. The persistent dominance of forest biomes at the Zoige Basin until ~2.7 Ma is in broad agreement with the model at 300–400 ppmv configuration.

We further performed a structural equation modelling (SEM) analysis and a series of vegetation simulations (Methods) to quantify the relative importance of global climate drivers (ice volume, $CO_2$ and summer insolation) which are probably involved in TP vegetation dynamics. The SEM results support the above explanations, showing that ice-boundary condition plays the most prominent role in long-term vegetation transformations (Fig. 5a). On orbital scales, summer insolation is shown to be the crucial driver of tree abundances directly and indirectly (Fig. 5a). However, the progressively reinforced impacts of global ice volume are clear in the SEM outputs (Extended Data Fig. 8). The results also highlight the secondary importance of atmospheric $CO_2$ on forest growth, which is consistent with new findings from tropical western Africa that reveal the lower relative importance of $CO_2$ compared to climate variability on trees over the last ~0.5 Myr (ref. 29). These suggest that the large amplitude of climate changes on both long-term and glacial–interglacial scales may have outweighed the impacts of $CO_2$ (ref. 29).

Additionally, vegetation simulations for MIS KM5C, 125 and 117 ka, which represent the mid-Pliocene, the last interglacial and the start of the last glacial, respectively, indicate that TP vegetation is strongly

influenced by climate conditions driven by insolation, ice volume and $CO_2$ concentrations (Methods; Fig. 5b). The persistent dominance of the forest biome in the Pliocene aligns well with the model results forced with the KM5C orbital parameter and 300–400 ppmv of $CO_2$. Furthermore, orbital configurations cause more substantial changes in forest and grassland cover than $CO_2$ concentrations under varying climate conditions in KM5C, 125 and 117 ka (Fig. 5b), underscoring the secondary role of $CO_2$. Interestingly, the impact of $CO_2$ on Zoige vegetation varies with its concentration and its effect on forest growth weakens further when $CO_2$ levels exceed ~300–400 ppmv, suggesting a possible saturation effect. These findings highlight the importance of considering the differing impacts of $CO_2$ on vegetation under various climate and $CO_2$ scenarios in future model-based projections.

These global drivers can influence eastern TP vegetation by modulating the monsoon climate. A stronger monsoon, characterized by warmer and wetter conditions, typically supports the expansion of tree populations. However, in the alpine Zoige region, vegetation dynamics within the meadow–conifer forest ecotone are primarily controlled by temperature[7,12,15,30], as moisture availability is generally sufficient due to the combination of low temperatures and moderate precipitation

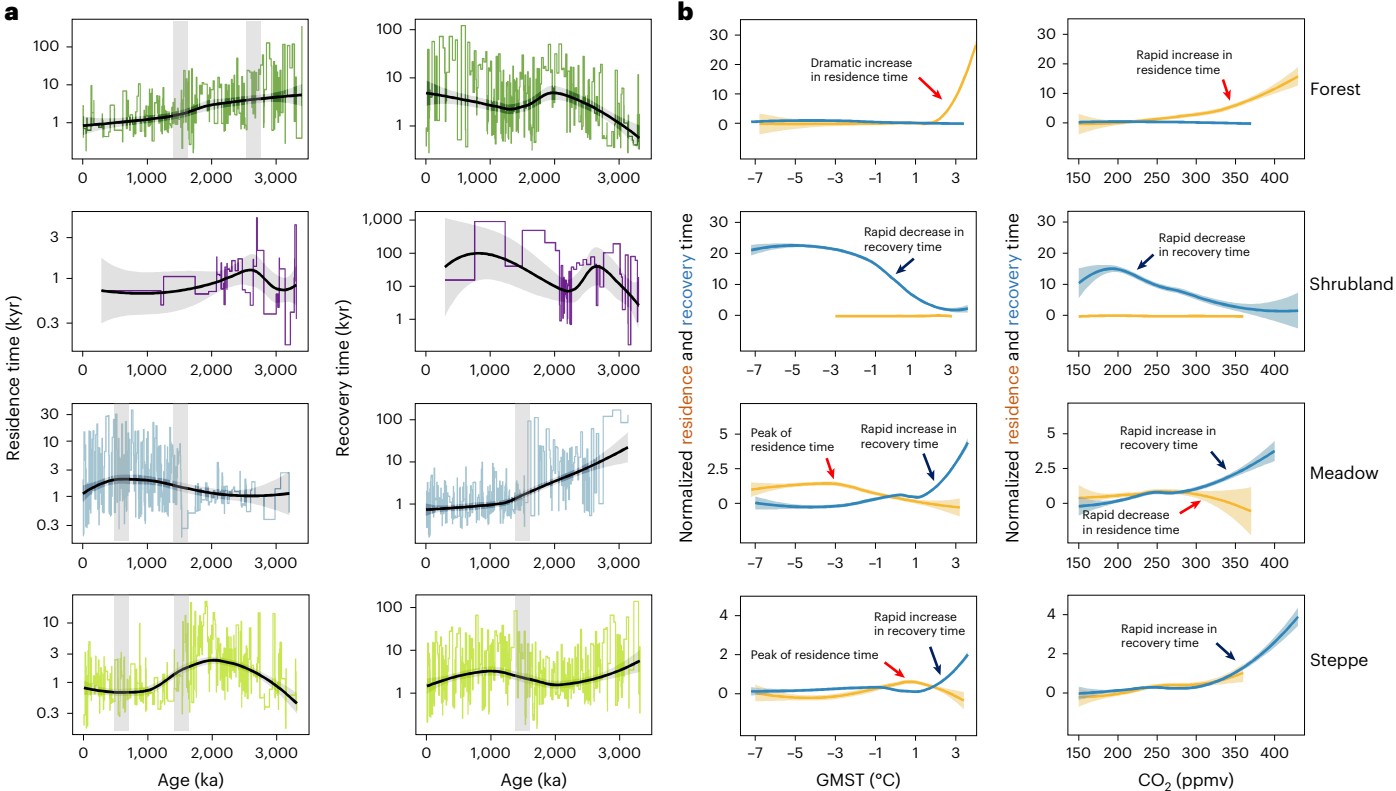

**Fig. 6 | Resilience changes of major megabiomes in the Zoige region and their relationships with global climate. a**, Residence time and recovery time (plotted on a logarithmic scale) reflecting the resilience of megabiomes. Their trends are illustrated by LOESS-fitted regression curves (bold black line), accompanied by 95% confidence intervals. Vertical grey bars denote periods during which substantial vegetation transformations occur. **b**, Relationships

between resilience and GMST as well as $CO_2$. Bold lines represent the LOESS-fitted regression curves of normalized residence time (yellow) and recovery time (blue), accompanied by 95% confidence intervals. Red and dark blue arrows mark the notable changes. The GMST values relative to PI are based on proxy-based reconstructions[20] and the $CO_2$ values are derived from simulations[19].

levels (Methods). This interpretation is supported by several lines of evidence: the distinct elevational distribution of modern vegetation (Fig. 1d); the predominance of forest, meadow and meadow-like steppe biomes (Extended Data Fig. 3a) and the strong correlation between AP% and the first axis of principal component analysis (PCA) on pollen data over the last 3.5 Myr (Extended Data Fig. 2e); and the close relationship between temperature and PCA axis 1 along with the overall lack of significance of precipitation in the significance tests of quantitative reconstructions for core ZB13-C2 (refs. 7,30). While temperature is a primary driver, drought stress would also have an impact on vegetation dynamics. Precipitation would further enhance forest growth during strong monsoon or interglacial intervals. Conversely, drought stress would also help to limit forest growth and alter grassland composition during weak monsoon or glacial intervals.

**Vegetation resilience and climate thresholds**

To explore quantitatively how these TP vegetation transformations and shifts responded to climate changes driven by global configurations, we estimated vegetation resilience by calculating the residence and recovery time of various biomes over the last 3.5 Myr (Methods). Higher/lower resilience is represented by longer/shorter residence time and shorter/longer recovery time. In general, the residence time of the dominant biomes of forest, meadow and steppe persist longer than that of shrubland and desert-steppe (Extended Data Fig. 9).

The resilience of the dominant biomes differs through time, in parallel with the long-term vegetation transformations. Resilience of forest decreased over the above-discussed four vegetation phases (3.5–2.73, 2.73–1.54, 1.54–0.62 and 0.62–0 Ma) (Fig. 6a and Extended

Data Fig. 9a,b), with the strongest resilience (mean residence time ~15 kyr and recovery time ~2 kyr) in the Pliocene and the weakest resilience (mean residence time ~1 kyr and recovery time ~18 kyr with highest value of >~100 kyr) for the last 0.62 Myr. Shrubland mostly occurs in the glacials of the mid-Pliocene and early Quaternary, with a mean residence time of ~1 kyr and recovery time of ~40–45 kyr. Steppe shows the greatest resilience in the glacials between 2.73 and 1.54 Ma, with a relatively high mean residence time (~3 kyr) and low mean recovery time (~3 kyr). The strong resilience of meadow since 1.54 Ma is mostly based on its long residence time (mean value ~3.5 kyr) and short recovery time (mean value <~1.5 kyr).

These resilience changes over the four phases correspond to a stepwise decrease of GMST[20] from median values of ~3, to 1.3, −1.7 and −3.4 °C relative to PI, as well as $CO_2$ decrease from ~300, to 250 and 220 ppmv (Methods) (Extended Data Figs. 6c and 9c). There is no regional temperature reconstruction available on the TP for the last 3.5 Myr; however, considering the amplification effect of albedo associated with high elevation[31,32], the magnitude of TP temperature decrease could be even larger than that of GMST.

In the mid-Pliocene, ~2–3 °C global warming would have raised the upper treeline several hundreds of metres, assuming a temperature lapse rate of ~0.62–0.65 °C per 100 m (refs. 33,34). This treeline migration rate would probably be also constrained by moisture stress, as many observations and a recent modelling study have revealed[35]. Nonetheless, the Zoige Basin (with an elevation of ~3,400 m a.s.l.) was therefore located in the central part of the conifer forest belt, taking into account the current upper elevational limit (~3,800–4,000 m) (ref. 12). This explains the enhanced forest resilience even in the glacials

and weak monsoon periods, as it took less time for the forest to recover from open vegetation. The stronger summer Asian monsoon[36] during this interval could further stabilize the forest belt. Furthermore, the high forest diversity as inferred from tree pollen richness (Extended Data Fig. 4b) as well as the higher *Tsuga* abundances and occasional occurrence of *Keteleeria* growing in warmer lower elevations, may have increased the resilience of forest ecosystem.

In the interval of 2.73–1.54 Ma, GMST was still above the PI level for both interglacials and glacials (Extended Data Fig. 6c,d). This explains the stronger resilience of shrubland and steppe compared to meadow in the glacials, as the major shrub and steppe taxa (*Hippophae*, Rosaceae, *Artemisia* and Amaranthaceae) prefer higher temperatures (particularly summer temperature) than those of meadow (Cyperaceae, Ranunculaceae and other major meadow taxa) (Extended Data Fig. 1c). This is also supported by modern regional pollen results showing that *Artemisia* and *Hippophae* have higher percentages at lower elevations of <~3,200–3,500 m (ref. 14). Suppressed precipitation in the glacials due to a weakening monsoon, together with stronger evaporation due to the relatively higher temperature than that of ~1.5–0 Ma, may well have caused lower effective moisture. This association of temperature and moisture may have promoted the resilience of shrubland and steppe, while restricting meadow expansion.

After ~1.54 Ma, GMST of all the glacials and most of the interglacials decreased to values lower than PI (Extended Data Fig. 6c,d). The modern meadow on the TP is optimally distributed in areas with a regional MAT of ~−1–0 °C and mostly in regions with MAT of ~−5–3 °C (Extended Data Fig. 1b). Recent studies suggest that TP temperature may have risen by ~1.5 °C since PI[37,38]. MAT at the Zoige Basin around PI is presumably <0 °C, considering the current value of ~1.4 °C. The median temperature after ~1.54 Ma should thus probably be <~−1.5 °C, which is ideal for meadow expansion. Accordingly, typical meadow components that favour cold climate (for example, Ranunculaceae, *Polygonum*, *Thalictrum* and various Asteraceae taxa) along with Cyperaceae increased in both glacials and some interglacials (Fig. 3). Enhanced diversity of herbs may have further maintained the long residence time and short recovery time of meadow (Extended Data Fig. 4b). Meanwhile, frequent millennial-scale climate changes in this interval[26] could facilitate the biome shift from forest to meadow even in interglacials, as it would take longer for forest than meadow to recover after frequent disturbance[39,40]. Consequently, a meadow-dominated ecosystem on the eastern TP was finally established after ~1.54 Ma, with a particularly sharp decrease of forest resilience after ~0.62 Ma, which can be explained by the longer migrational distance for tree populations to seek refugial locations at increasingly lower elevations with the emergence of lengthy and extreme glacials[17].

Moreover, we identify the climate thresholds for vegetation resilience shifts by analysing their relationship with temperature and $CO_2$ regardless of time intervals and glacial–interglacial cycles (Fig. 6b and Extended Data Fig. 10). The results demonstrate that ~2–3 °C warming relative to PI and ~350 ppmv of $CO_2$ are major tipping points for a substantial increase in forest resilience. Shrubland and steppe have the strongest resilience under ~1–2 °C warming, with the recovery time increasing rapidly with $CO_2$ >350 ppmv. Conversely, once the thresholds of ~2–3 °C warming and ~300 ppmv of $CO_2$ are crossed, meadow takes much longer to recover. The GMST rate-of-change shows no straightforward impact on resilience of forest and steppe, but reveals a clear impact on meadow resilience mostly by shortening recovery time once it crosses the value of ~0.5 °C kyr⁻¹ for decrease (Extended Data Fig. 10), although the insufficient resolution of GMST may yield uncertainty in the estimates of this correlation. These estimated temperature thresholds further quantitatively support the above explanations.

## Future implications

These new lines of evidence provide a reversed analogue of TP vegetation response to projected future global warming. Elevated temperature would cause the meadow to lose, ultimately, its resilience and transform it back to shrubland/steppe and further to forest, if global warming exceeds ~1–2 and ~2–3 °C relative to PI (similar to the projected values by the end of this century under shared socioeconomic pathways (SSP) 1-2.6 and 2-4.5)[41]. Recent studies indicate that increased precipitation with warming on the TP may promote modern meadow to some degree[42]. However, the Zoige Basin palaeorecord highlights that the association of a warmer and stronger monsoon in the Pliocene and early Quaternary outweighed this precipitation effect on meadow when temperature reached certain thresholds. Meanwhile, the elevated $CO_2$ concentrations along with increasing transient human disturbances in the future[43] may also expedite this imminent steppification process. Nevertheless, our results indicate that TP vegetation is highly sensitive to temperature change and suggest that, without major reductions in greenhouse gas emissions to the atmosphere, the current meadow is at risk of major transformation. As meadows constitute >60% of the alpine grassland area on the TP, they play a vital role in water conservation, soil preservation, carbon sequestration, biodiversity protection and climate regulation at both regional and global scales[2,42]. Consequently, these ecosystem transformations would greatly impact the hydroclimate, diversity, ecosystem stability and ecological services of the TP and a large area of Asia.

## Methods

### Regional settings

The Zoige Basin is a low-relief tectonic basin (32° 10′–34° 10′ N, 101° 45′–103° 25′ E, ~3,350–3,450 m a.s.l) on the eastern TP (Fig. 1a–c). A huge lake occupied the basin during the Pleistocene, until it was finally drained ~30–20 ka when the Yellow River cut through the mountain barrier to the east[44].

The Zoige Basin is currently covered by subalpine meadows dominated by *Kobresia* spp. and other Cyperaceae taxa. A distinct elevational vegetation gradient occurs in the mountains surrounding the basin (Fig. 1d and Supplementary Table 2)[7,12]. In the mountains south and east of the basin, <~3,000 m there is the zone of montane conifer and broadleaf mixed forest, which is distributed down to ~2,000 m. Conifer forest, dominated by various species of *Picea* and *Abies* with some broadleaf trees such as *Betula*, *Quercus* and *Acer*, is mainly located at ~3,000–4,000 m. Scattered evergreen sclerophyllous *Quercus* forest is found on south-facing slopes <3,800 m. The zone between ~3,000 and 4,400 m is occupied by subalpine and alpine shrubland and meadow belts with a boundary at ~4,000 m. Various *Kobresia* spp. prevail in the two belts, but different species dominate them. *K. pygmaea* characterizes alpine shrubland and meadow, while *K. setschwanensis* mainly occurs in subalpine shrubland and meadow. Grasses are also important components in these two vegetation belts, primarily including *Elymus nutans*, *E. burchan-buddae*, *Stipa capillacea*, *S. aliena* and *S. purpurea*. In addition, diverse shrubs (for example, *Rhododendron* spp., *Sibiraea angustata*, *Spiraea schneideriana*, *Salix sclerophylla*, *Dasiphora fruticosa* and *Hippophae tibetana*) and forbs (*Anemone* spp., *Argentina anserina*, *Saussurea* spp., *Anaphalis flavescens*, *Leontopodium junpeianum*, *Bistorta vivipara* and *B. macrophylla*) can be frequently observed in the subalpine and alpine shrubland and meadow belts[13,45]. Alpine sparse vegetation occurs from >~4,400 to 4,500 m. The peaks >~4,600 m are covered by ice and snow. Similar vegetation zones can be found on mountains to the north of the basin, although their elevational limits are generally ~200–300 m lower[12].

### Seismic survey, drilling operation and sediment lithology

The seismic survey was conducted in the sedimentary centre of the palaeolake Zoige using Tromino 3G seismometers[7] to assist the drilling site determination. Deep drilling was performed in June–August 2019. A 1,084.67-m core reaching the basal rock was retrieved, with a recovery of 95.4%. No obvious sampling gaps were detected, except a hiatus from 2,964 to 2,934 ka.

The lithology of the sediment core (Fig. 2a) mainly consists of clay, silty-clay and clayey-silt, except for two fluvial sandy layers (10.18 and 9.14 m thick, respectively). The top 583 m of the core is identical to the previously drilled core ZB13-C2 covering the last ~1.72 Myr (ref. 7) and the bottom section is dominated by fine bluish-grey clay, indicating a lacustrine origin. The persistent presence of aquatic pollen (Fig. 3a) and freshwater Ostracoda taxa (Supplementary Table 3) further support the freshwater nature of the lake deposits.

The high recovery rate of core drilling, the predominance of lacustrine-originated clay sediments, straightforward correlation with orbital parameters and the stable sedimentation rate (Fig. 2b,c) collectively indicate the near-continuous nature of the sediments, suggesting the absence of substantial hiatuses.

## Palaeomagnetic measurements

A total of 3,877 samples of 2-cm-edge cubes from core ZB19-C1 were used for palaeomagnetic measurements. Before demagnetization, anisotropy of magnetic susceptibility (AMS) measurements were performed using a KLY-4s Kappabridge (Agico). AMS results indicate that the sediments are generally undisturbed since their deposition and suitable for palaeomagnetic analysis. Rock magnetic investigations show that the main magnetic carrier is low coercivity ferrimagnetic mineral dominated by magnetite, suggesting that stepwise alternating field (AF) demagnetization is a suitable approach[7].

Remanence measurements were then made using a 2G-760 cryogenic superconducting magnetometer in a magnetically shielded space (<300 nT) at Paleomagnetism and Geochronology Laboratory. All the 3,877 samples were subjected to stepwise AF demagnetization up to 75 mT at 2.5–10-mT intervals. A total of 2,372 samples gave reliable characteristic remanent magnetization (ChRM) directions. For the samples whose AF demagnetization results were not satisfactory, thermal demagnetization was further measured using a TD-48 thermal demagnetizer. They were stepwise heated to 680 °C at 10–50 °C increments. Both AF and thermal demagnetizations can effectively remove viscous components of magnetization after 15–20 mT or 250 °C, respectively. A total of 72% samples yielded stable and reliable ChRM directions.

The recognized magnetic polarities were then correlated with the geomagnetic polarity timescale (GPTS)[46] (Supplementary Table 1 and Fig. 2a). Two main geomagnetic reversals, the Matuyama/Brunhes and Gauss/Matuyama boundaries) were clearly identified. The subchrons in the Matuyama reverse chron, including Jaramillo, Cobb Mountain, Olduvai and Reunion and the reverse subchrons in the Gauss chron, including the Kaena and the Mammoth were also clearly identified. The geomagnetic polarity pattern of the upper part of ZB19-C1 is perfectly consistent with ZB13-C2 (ref. 7).

## Pollen analysis

A total of 5,000 samples were analysed from the combined cores ZB13-C2 and ZB19-C1. For the new core ZB19-C1, 2,088 pollen subsamples of ~2 cm³ were collected at intervals of ~40 cm, except for the Pliocene section, where samples were taken at 20–10 cm intervals to capture detailed pollen and vegetation changes during this critical period for the data-model comparison project[4]. For core ZB13-C2, a total of 2,912 samples were analysed, including 2,787 previously published samples[7] and 125 newly analysed samples to achieve higher resolution data for certain interglacial periods.

All the samples were processed following standard procedures[47], including HCl, KOH, HF, heavy liquid flotation and acetolysis treatments, as well as fine sieving to remove clay-sized particles. A tablet containing a known quantity of *Lycopodium* spores (Lund batch no. 1031, 20,848 grains per tablet) was added at the beginning of processing to estimate pollen concentrations[48]. Pollen identifications were conducted using published literature[49–51] on Chinese pollen morphology and reference collections. At least 300 terrestrial pollen grains were counted for each sample.

We uniformly adopted the widely accepted nomenclature for flowering plant families as proposed by the Angiosperm Phylogeny Group[52]. Chenopodiaceae is merged into the family Amaranthaceae. The Amaranthaceae pollen identified in this study specifically belongs to its subfamily Chenopodioideae, which is characteristic of a highly continental climate with cold winter and dry summers[53].

The pollen percentages were calculated on the basis of the sum of the terrestrial pollen taxa. Pollen influx was calculated by multiplying pollen concentration and sedimentation rate derived from the dating levels and ETP tuning points (Fig. 2c and Supplementary Table 1). Tilia 1.7.16 software[54] was used for the calculation of percentages, drawing initial pollen diagrams and zonation division using the CONISS[55] function.

To ensure consistency among analysts, we dedicated our efforts on meticulous methodology before initiating pollen analysis, which included standardizing the pretreatment protocol, creating reference pollen slides and establishing uniform counting rules. Additionally, we conducted several rounds of cross-checking to verify consistency between analysts. The results for the same sample show differences falling within an acceptable range (for example, most AP% difference range being <3%). Moreover, the inferred boundaries of vegetation transformations at ~2.73, 1.54 or 0.62 Ma do not coincide with transitions between different analysts. Furthermore, AP% in core ZB13-C2 show coherent changes with X-ray fluorescence-based Rb/Sr ratio and carbonate percentages across timescales[7]. All these support our conclusion that analytical differences did not introduce any artefacts of vegetation shifts.

## Establishment of age–depth model and composite record

First-order tie points are provided by the magnetostratigraphic results (Supplementary Table 1), which were used in constructing an initial age model. As revealed by ref. 7, spectral and wavelet analysis of AP% show clear cycles in depth domain, suggesting possible E, T and P powers[7]. AP% were then tuned to the ETP (Fig. 2b and Supplementary Table 1), that is generated by normalizing and averaging variations in orbital E, T and P[16] and the LR04 benthic $\delta^{18}$O stack record[17]. Comparison of the orbital tuning and land–sea correlation age models shows close similarities. To avoid any circularity, the final age was established on the basis of the ETP correlation control points when exploring high-latitude ice-sheet forcing on orbital-scale vegetation changes. The basal age for both models dated to 3.5 Ma.

AP% for the top 583 m of core ZB19-C1 were aligned to that of core ZB13-C2 according to the perfect correlation of AP% in the overlapped section of ~1.72–1.58 Ma of the two cores. Note that the bottom age of ZB13-C2 was adjusted from 1.74 to 1.72 Ma on the basis of the Olduvai reversal identified in core ZB19-C1. Combination of the ZB13-C2 and ZB19-C1 data provided a uniquely continuous high-resolution palynological archive (with an average ~700-yr time resolution) of long-term, orbital and millennial-scale pollen and vegetation variability for the last 3.5 Myr (Fig. 3).

## Biome reconstruction

The biomization method[56] was applied to reconstruct palaeovegetation. Pollen taxa were assigned to one or more plant functional types (PFTs), then characteristic PFTs were used to define principal vegetation types (biomes). A modified scheme for PFTs and biomes of China by ref. 57 was used, in which pollen taxa were assigned into PFTs and PFTs into biomes (Supplementary Table 4) on the basis of the modern ecology, bioclimatic tolerance and spatial distribution of pollen-producing plants. The biome score calculation was performed following ref. 56. The biome with the highest affinity score or the one defined by fewer PFTs (when scores of several biomes are equal) was assigned to be the dominant biome for each pollen assemblage.

The biomes were then combined into five megabiome types which represented the vegetation types of the eastern TP (forest, meadow,

steppe, shrubland and desert-steppe), according to the equivalents of them in China's vegetation classification[57] and the pollen taxa related to them (Supplementary Table 4). All forest biome types (Supplementary Table 4) were merged into the forest megabiome. Temperate grassland and temperate xerophytic shrubland, which had similar pollen taxa composition, were grouped into steppe. Five tundra biomes were assigned to shrubland and meadow megabiomes, as they represent the subalpine and alpine shrubland and meadow vegetation on the TP[57]. Erect dwarf-shrub tundra, low- and high-shrub tundra and prostrate dwarf-shrub tundra were grouped into shrubland, because they were defined as shrubby vegetation and consisted of various shrub pollen taxa (Supplementary Table 4). Graminoid and forb tundra and cushion-forb tundra had similar pollen taxa composition and were characterized by typical meadow elements, so they were combined into the meadow megabiome.

The high score of meadow in the samples assigned to shrubland and steppe biomes (shown by high ratios of meadow to shrubland or steppe; Extended Data Fig. 3a), particularly for the last 1.5 Myr, suggest that shrubland and steppe biomes are meadow-like in most cases. Although 23 samples were assigned to the desert biome (Extended Data Fig. 3a), their pollen assemblages did not show typical characteristics of a desert or even desert-steppe pollen spectrum with dominant Amaranthaceae and *Artemisia* as well as xerophilous elements (for example, *Ephedra*, *Nitraria*, *Tamarix* and Zygophyllaceae) in arid and semi-arid regions[58]. Instead, they have relatively high percentages of Cyperaceae and Poaceae pollen. These desert samples were therefore assigned to desert-steppe vegetation (also meadow-like).

Fossil spectra with very low pollen influx under cold/dry climates may be potentially dominated by long-distance transported arboreal pollen, leading to the misclassification of vegetation types. To address this issue, we calculated pollen influx (Extended Data Fig. 3b) to validate the biomization results. Several lines of evidence suggest that the influence of extra-source arboreal pollen on our vegetation type reconstructions is minimal. First, pollen influx for individual taxa generally aligns well with pollen percentages. Second, AP% exhibit a clear positive relationship with total pollen influx and high AP% values are not associated with extremely low concentrations or influx. Third, the samples classified as belonging to the desert-steppe biome do not show very low pollen concentrations or influx. Their distinct vegetation composition and structure contrasting that in arid and semi-arid regions[58] further support the conclusion that vegetation cover was denser and less affected by this problem, even during stadials of glacial periods. Given that sparse vegetation is currently observed at elevations >4,400 m (ref. 12), where temperatures are ~6.5 °C lower than at our study site due to the ~1,000-m elevation difference, this assumption is reasonable. Moreover, the coherent changes in AP% with the Rb/Sr ratio and carbonate content from core ZB13-C2 (ref. 7) also confirm the robustness of the vegetation reconstruction.

### Diversity estimation
We used pollen richness and evenness to measure palynological diversity. We applied rarefaction analysis to calculate the palynological richness index, which standardizes the pollen count size to a constant number of grains in each sample for the entire core, to make the comparisons of pollen richness between different samples realistic and meaningful[59,60]. Modern pollen diversity investigations have demonstrated a positive correlation between rarefaction-based palynological diversity and floristic diversity[60–62]. Because rarefaction does not allow extrapolation[59], the standardization constant number must be equal to or less than the minimum count size. A pollen sum of 300 grains was thus adopted for standardization. Pollen evenness was evaluated by calculating the Pielou index[63].

The representation of palynological diversity in relation to vegetation diversity can be influenced by several factors, such as vegetation openness[64], count size[65] and the taxonomic precision of pollen identification[60]. However, palynological diversity can remain comparable within a single core over an extended time series[60,66,67], provided that count size, taxonomic precision and laboratory protocols stay consistent over time. Among these factors, vegetation openness may introduce greater uncertainties in long time series compared to others. In our study region, previous research has shown that vegetation openness plays a secondary role[67]. Nevertheless, to account for the potential influence of openness on total diversity, we also separately analysed the diversities of tree taxa and shrub/herb taxa.

### Vegetation resilience estimation
We analysed vegetation resilience by calculating residence and recovery time. The residence time of each megabiome within the core was calculated using the midpoint between the two adjacent fossil samples where there is a biome transition, and the recovery time was estimated as the interval between the last occurrence of the megabiome to the first reoccurrence of the megabiome[40]. If the same type of biome does not re-occur, we designated the biome type as not recovering. The changes of resilience over time for megabiomes were separately assessed (Fig. 6 and Extended Data Fig. 10), except for the desert-steppe biome which had a small sample size (23 samples). Relationships between resilience and climate were analysed for different megabiomes to explore the possible impact of global climate on TP vegetation resilience, including GMST relative to the PI[20] and simulated $CO_2$ data[19]. Resilience was also compared with rate-of-change of temperature, which is calculated as the difference between two consecutive data of GMST. To better illustrate the trends of resilience, locally estimated scatterplot smoothing (LOESS) regression was used to fit smooth curves for resilience changes on the timeline and climate gradient.

### Time-series analysis
To track the time-varying amplitude of orbital and suborbital periods, we calculated the continuous wavelet transform using Acycle[68] (Fig. 4b). Before the analysis, the data of AP% were resampled at equally spaced 1-kyr intervals and detrended. The statistical significance of the wavelet power was tested relative to a red-noise background power spectrum.

### The detection of regime shifts and ordination analysis
Overall, CONISS based on all pollen data identified distinctive pollen composition shifts at 2.73, 1.54 and 0.62 Ma (Fig. 3a). Multivariate regression tree, a hierarchical constrained clustering technique, was also performed on fossil pollen data with age as the external variable to explore major change points of pollen composition[69]. These two approaches show consistent results.

We further conducted recurrence analysis[70] in the time domain to identify transitions between different regime states (Extended Data Fig. 2a). Analysis was performed on the non-detrended AP% records. To better quantify the recurrence plot, 'determinism' (DET)[71] was introduced to assess the fraction of recurrence points that form diagonal lines with respect to all recurrence points. This parameter quantifies the predictability of dynamics in the state of a system. Predictability estimates the stochastic (unpredictable) versus the deterministic (predictable) nature. DET values near zero correspond to unpredictable dynamics, whereas large values indicate predictable dynamics and tipping points.

Ordination techniques were used to explore relationships among pollen assemblages. Only those pollen taxa with a percentage >2% in at least one sample were included in ordination analysis, and they were standardized by a square-root transformation. A preliminary detrended correspondence analysis indicated that the length of standard deviation unit on the first axis was 2.49, suggesting that a linear method is appropriate to analyse this dataset. Therefore, PCA was applied to the fossil pollen data (Extended Data Fig. 2c,d).

## Rate-of-change analysis

Rate-of-change (RoC) analyses were used to estimate the magnitude of the LR04 $\delta^{18}O$ stack and GMST change per unit time to investigate its impact on vegetation shifts (Extended Data Fig. 6b,c). First, a smooth generalized additive model (GAM) was used to estimate trends of the $z$-score transformed time series, and restricted maximum likelihood was used for smoothness selection[72]. Second, we calculated the first derivative of the estimated trend to represent the RoC using finite differences[73,74]. Finally, the periods of substantial change were identified where the simultaneous confidence interval of the first derivative of the GAM function moved away from zero[73].

## Structural equation modelling

SEM[75,76] was applied to assess the relative importance of TP climate drivers. We followed the detailed procedures described in ref. 29. In our study, SEM analysis is an a priori conceptual model which describes, schematically, how the three drivers (summer insolation, ice volume and $CO_2$ concentration) may interact to determine temperature inferred from AP%. Summer insolation is represented by mean June insolation at 30° N (ref. 16). Ice volume is represented by the LR04 $\delta^{18}O$ stack[17] and $CO_2$ data are derived from simulated data[19]. The conceptual model is based on our understanding of the global factors influencing vegetation dynamics on the eastern TP[7,12].

Before running the SEM, three data transformations were performed. First, we multiplied the LR04 $\delta^{18}O$ stack data by −1 as low values for $\delta^{18}O$ represent high ice volume. After multiplication, low values mean low ice volume. Second, we perform bandpass filtering on AP% data with a time window of 13–250 kyr, to retrieve orbital signals. Third, we scaled all the predictors to a mean of 0 and standard deviation of 1. All data were then aligned to 1-kyr resolution by using a linear interpolation. To explore the dynamic impact of the three drivers across the four vegetation phases, we performed the analysis individually for each interval.

Besides the hypothesized direct and indirect relationships, we also added correlations between the measured variables: ice volume–$CO_2$, insolation–ice volume and insolation–$CO_2$. Spearman rank correlations are given for all pairs of variables in our SEM for the dataset (Extended Data Fig. 8 and Supplementary Table 5), to assist in assuming prior models between variables.

Aligning the differing chronologies across various records may introduce certain uncertainties. The age differences between the LR04 $\delta^{18}O$ stack[17] and simulated $CO_2$ records[19] are negligible, whereas they differ by several thousand years when compared to summer insolation[16]. Additionally, the AP% record, with its age tuned to the ETP, exhibits a phase difference of up to several thousand years when compared to the LR04 stack and global atmospheric $CO_2$ records. To address these discrepancies, we performed additional SEM analyses using an alternative chronology obtained by tuning AP% to the LR04 stack record. The results consistently underscore the fundamental role of insolation on the orbital scale, the growing influence of global ice volume over long-term change and the secondary role of $CO_2$, aligning with the pattern illustrated in Extended Data Fig. 8. They reinforce the robustness of our overarching conclusions, despite potential uncertainties arising from age discrepancies among the parameters.

## Vegetation simulations

We used the LPJ-GUESS vegetation model, driven by climatology from the iLOVECLIM and NorESM-L climate models (Fig. 5b). LPJ-GUESS[77,78] uses identical biophysical and physiological process parameterizations to BIOME3 (ref. 79). Additionally, LPJ-GUESS incorporates dynamic representations of establishment, mortality, growth, carbon allocation, plant allometry and dynamic competition among 11 PFTs. This model has been widely used to investigate past and future vegetation dynamics[78,80], as well as to explore their interactions with climate and $CO_2$ concentrations[81,82]. The iLOVECLIM[83,84] and NorESM-L[85,86] climate

models have been successfully applied in the previous studies for different periods[87–92]. By combining the LPJ-GUESS vegetation model with climatology from these two climate models, we analysed vegetation dynamics under specific climate and $CO_2$ conditions.

We conducted a series of sensitivity experiments. The simulations comprise 12 combinations of input conditions, including three sets of monthly climate conditions and four $CO_2$ levels. The climate conditions represent a cold boreal summer orbit climate (117 ka), a warm boreal summer orbit climate (125 ka) and a Pliocene climate (KM5c). The four $CO_2$ levels are 200, 300, 400 and 560 ppmv. Eight vegetation simulations were run using LPJ-GUESS with cold (glacial) and warm (interglacial) boreal summer orbit climate forcing from iLOVECLIM under the four $CO_2$ scenarios. Additionally, four simulations were performed with LPJ-GUESS using the Pliocene (KM5c) climate forcing from NorESM-L, also under the four $CO_2$ scenarios. LPJ-GUESS was run 'off-line', using monthly mean temperature, precipitation and cloud cover from climate experiments, based on the average over the last 100 years of simulations. Each vegetation simulation began with a 1,000-yr spin-up phase and continued to equilibrium, with the vegetation averages calculated from the last 30 yr of the simulations.

The orbital parameters for the climate experiments were varied to represent (1) cold (glacial) boreal summers, (2) warm (interglacial) boreal summers and (3) Pliocene climate conditions. The selected values for orbital eccentricity, obliquity and the longitude of perihelion (the prograde angle between the vernal equinox and perihelion) at 117 ka (cold boreal summer orbit) and 125 ka (warm boreal summer orbit) follow the PMIP4 experimental protocol[93,94]. These orbits are considered broadly representative of conditions favourable (cold) or unfavourable (warm) for ice growth during both the Pliocene and late Pleistocene. For the Pliocene, the climate conditions were based on a time slice of 3.205 Myr, centred on an interglacial peak (MIS KM5c), which represents the warm conditions of the Pliocene[4].

## Climate tolerance of meadow and major pollen taxa

Probability density functions of climate variables were calculated to evaluate the favourable climate conditions for meadows on the TP (Extended Data Fig. 1b), based on modern meadow distributions retrieved from the 1:1,000,000 vegetation map of China[95] and their corresponding climate data extracted from the 30-s historical climate data (1970–2000) of WorldClim v.2.1 (ref. 96).

Climate optima and tolerances of the major taxa in fossil pollen spectra of the Zoige Basin were estimated by applying a weighted averaging method[97] (Extended Data Fig. 1c) based on modern pollen[30] and climate data[96] from 1,448 sites on the TP.

## Reporting summary

Further information on research design is available in the Nature Portfolio Reporting Summary linked to this article.

## Data availability

The data that support the findings of this study are included in the paper and/or the additional information. They are also available via figshare at https://doi.org/10.6084/m9.figshare.27966036 (ref. 98). Source data are provided with this paper.

## Code availability

All codes are derived from publicly accessible sources and no custom code was developed. Recurrence analysis was performed in Matlab using the CRP Toolbox[70]. The following analyses were conducted in R v.4.3.2 (ref. 99): Biomization, http://zenodo.org/records/7523423 (ref. 100); pollen diversity and ordination analyses, the vegan 2.6-6.1 package[101]; multivariate regression tree, the mvpart 1.6-2 package[69]; fitting GAM and rate of change, the mgcv 1.9-1 package[72] and gratia 0.9.2 package[74]; SEM, the lavaan 0.6-19 package[102].

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

## Acknowledgements

We gratefully acknowledge D. Za for logistical support and the villages (Nenwa and Maiqi), which hosted our fieldwork. We thank X. Sun, Y. Tang, S. Sun, L. Chen, C. Chen, Y. Fu, Z. He and R. Geng (field assistance); and H. Hooghiemstra (sharing Bogotá pollen data with interpretation). P. C. Tzedakis, L. Y. Tang, N. Q. Wu, F. J. Li, F. H. Chen and S. M. Wang are acknowledged for discussions. Financial support of this research was provided by National Key Research and Development Program of China (no. 2022YFF0801501 to Y.Z.) and National Natural Science Foundation of China (no. 42488201 to Z.G.).

## Author contributions

Y.Z. initiated and designed the study. F.Q., Q.L., Q.C., C.L., W.R., W.Z., M.C., H.L., Y.C., H.Y., H.Q. and Y.Z. undertook the fieldwork. Laboratory analyses were conducted by F.Q., Y.C., C.L., Q.C., Q.L., H.L., Z.Z., H.W., Y.L., Y.Z., H.Y., Y.K., J.Z., W.Z., H.Q. and H.X. J.G. and C.D. supervised the palaeomagnetism analysis. Y.Z. constructed the age model. F.Q., C.L., Y.Z. and H.L. performed biomization, numerical analysis or modelling simulation. The paper was written by Y.Z. with contributions from all co-authors.

## Competing interests

The authors declare no competing interests.

## Additional information

**Extended data** is available for this paper at https://doi.org/10.1038/s41559-025-02743-2.

**Correspondence and requests for materials** should be addressed to Yan Zhao.

¹Key Laboratory of Land Surface Pattern and Simulation, Institute of Geographic Sciences and Natural Resources Research, Chinese Academy of Sciences, Beijing, China. ²University of Chinese Academy of Sciences, Beijing, China. ³Department of Biological Sciences and Bjerknes Centre for Climate Research, University of Bergen, Bergen, Norway. ⁴Environmental Change Research Centre, Department of Geography, University College London, London, UK. ⁵School of Land Science and Space Planning, Hebei GEO University, Shijiazhuang, China. ⁶School of Geographical Science, Nantong University, Nantong, China. ⁷College of Geography and Tourism, Hengyang Normal University, Hengyang, China. ⁸Key Laboratory of Cenozoic Geology

and Environment, Institute of Geology and Geophysics, Chinese Academy of Sciences, Beijing, China. [9]Key Laboratory of Vertebrate Evolution and Human Origins, Institute of Vertebrate Paleontology and Paleoanthropology, Chinese Academy of Sciences, Beijing, China. [10]Hebei Normal University of Science & Technology, Qinhuangdao, China. [11]Jiangxi Provincial Key Laboratory of Wetland Plant Resources Conservation and Utilization, Lushan Botanical Garden, Jiangxi Province and Chinese Academy of Sciences, Jiujiang, China. [12]College of Earth and Environmental Sciences, Lanzhou University, Lanzhou, China. [13]Institute of Geomechanics, Chinese Academy of Geological Sciences, Beijing, China. [14]Qinghai Institute of Salt Lakes, Chinese Academy of Sciences, Xining, China. [15]School of Geography and Remote Sensing, Guangzhou University, Guangzhou, China. [16]Institute of Carbon Neutrality, Key Laboratory for Earth Surface Processes of the Ministry of Education, College of Urban and Environmental Sciences, Peking University, Beijing, China. ✉e-mail: zhaoyan@igsnrr.ac.cn

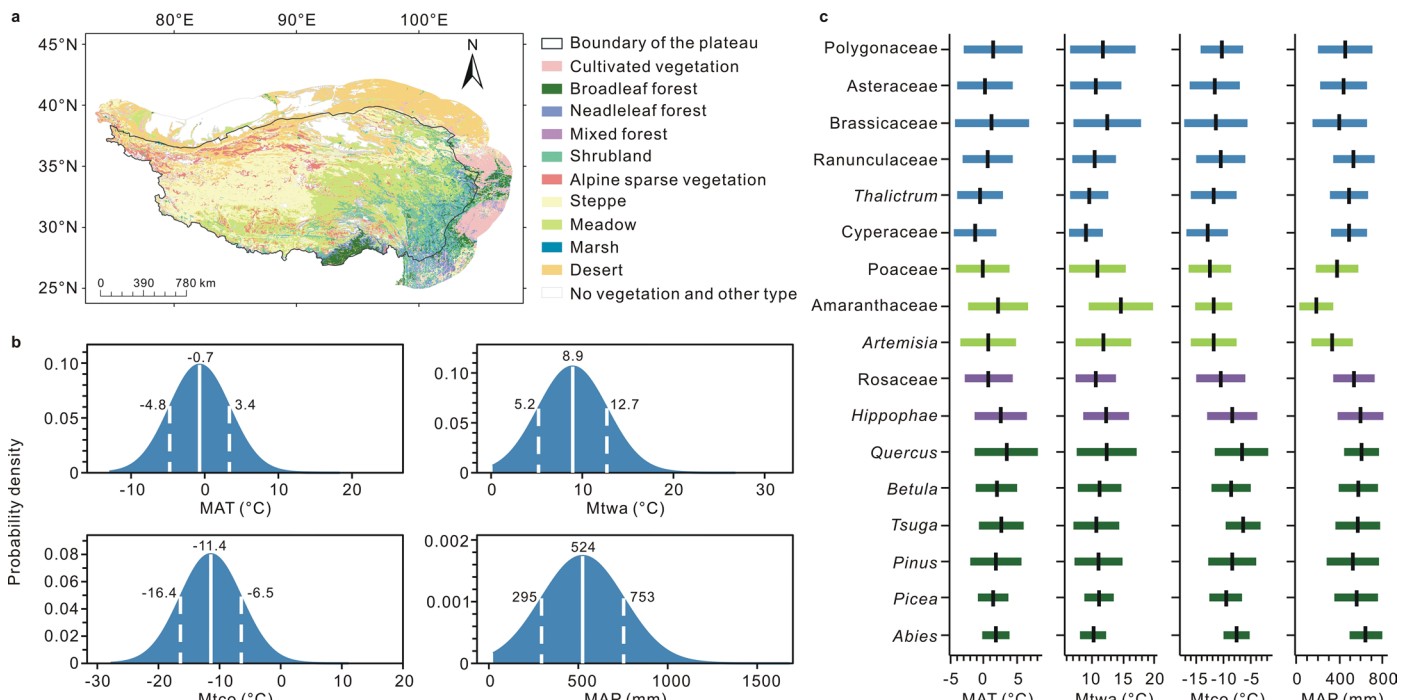

**Extended Data Fig. 1 | Modern vegetation distribution and climate envelope for meadow and major pollen taxa on the TP. a,** Modern vegetation distribution retrieved from the 1:1,000,000 vegetation map of China[95] (https://www.ncdc.ac.cn/portal/metadata/20d2728d-8845-4a8b-a546-5f4f50fb036d). Black line delineates the TP region. **b,** Probability density functions (PDF) of meadow occurrence vs. mean annual temperature (MAT), mean temperature of the warmest month (Mtwa), mean temperature of the coldest month (Mtco), and mean annual precipitation (MAP). The white solid lines and the numbers above them indicate the mean values of the PDFs, while the white dashed lines and the numbers near the top of them indicated the ±1 σ ranges. An optimal MAT of ~−5–3 °C favors meadow expansion. **c,** Climate optima (black squares) and tolerances (colored segments) of major pollen taxa. The color of each segment signifies the importance of the taxon within a particular vegetation type: dark green for forest, purple for shrubland, blue for meadow, and pale green for steppe.

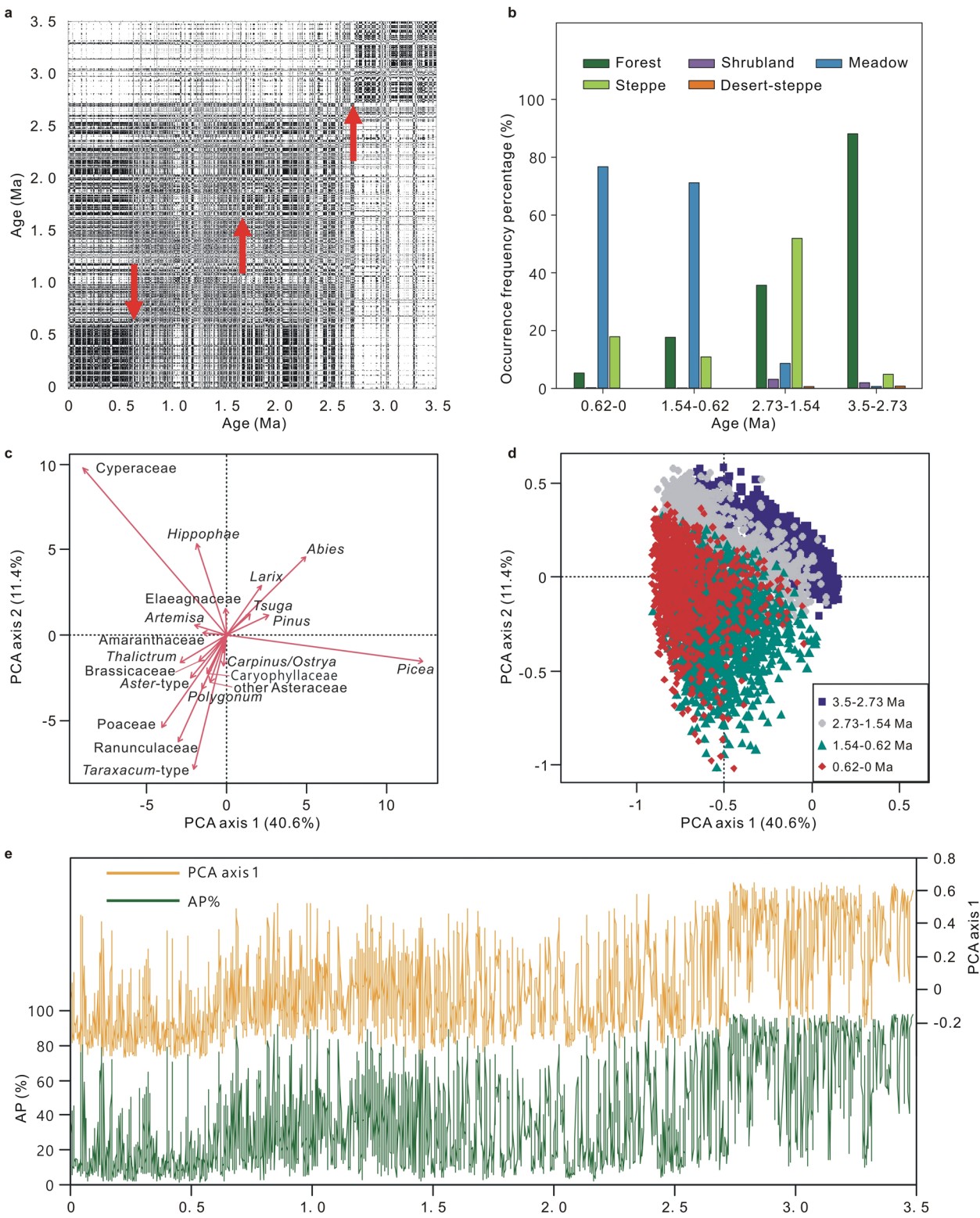

**Extended Data Fig. 2 | Vegetation regime shifts in the Zoige region revealed by a Recurrence Analysis (RA), biome occurrence frequency assessment, and principal component analysis (PCA). a**, RA results. Analysis was performed on the un-detrended AP% data. Determinism (DET) values close to zero correspond to unpredictable dynamics, whereas large values indicate predictable dynamics. The red arrows indicate three system shifts at -2.73, -1.54, and 0.62 Ma. **b**, Occurrence frequency percentage of major biomes for the intervals of 0.62–0, 1.54–0.62, 2.73–1.54, and 3.5–2.73 Ma. The percentage was estimated by 100*total residence time/interval span. **c**, **d**, PCA plots based on major pollen taxa and pollen assemblages of the four pollen zones (3.5–2.73, 2.73–1.54, 1.54–0.62, and 0.62–0 Ma), respectively. Distribution of pollen samples on the PCA plot indicates that pollen assemblages of the four pollen zones have clear distinctions, although some similarities are also observed among pollen zones. **e**, PCA sample scores on axis 1 versus AP%. They show coherent changes.

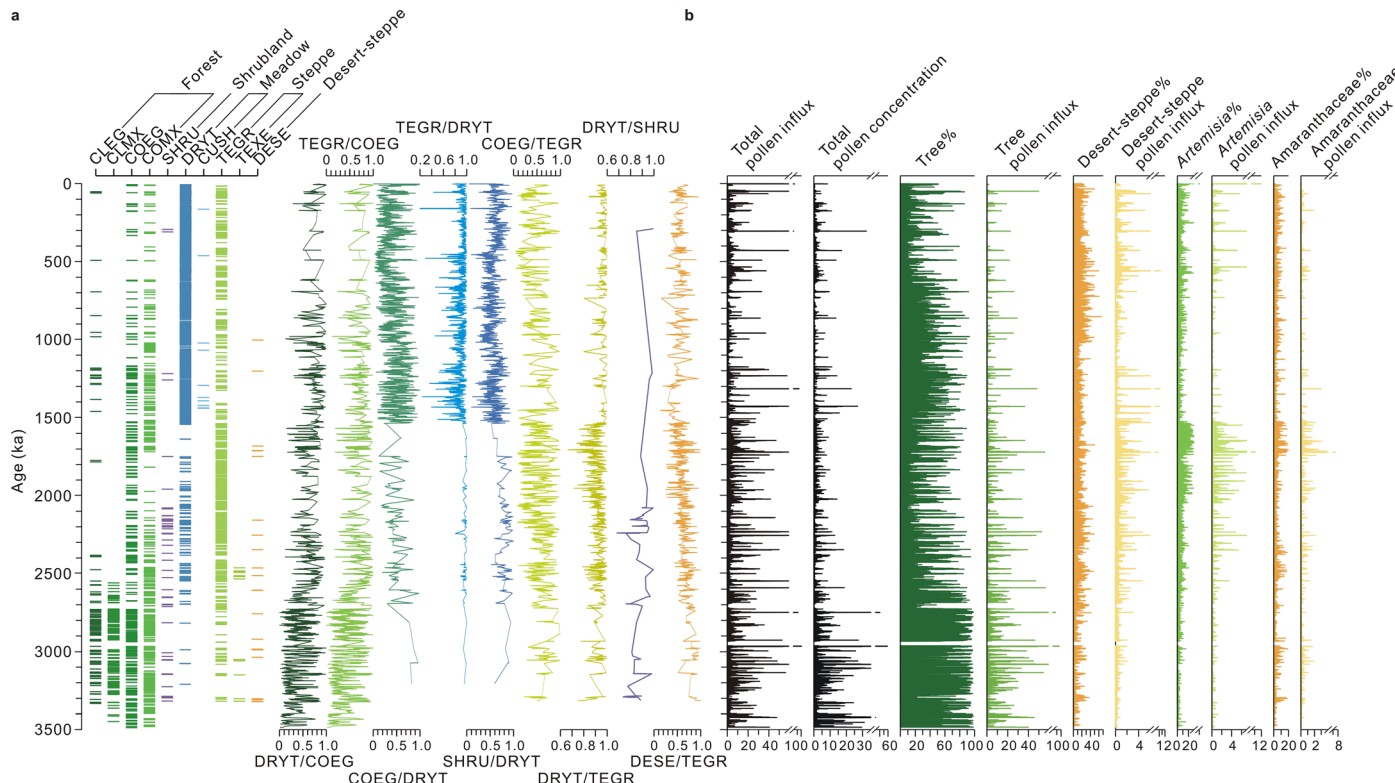

**Extended Data Fig. 3 | Reconstructed biomes and ratios of different biomes (a), and pollen influx (b) from the Zoige Basin core.** Biome ratio in panel **a** indicates the relative importance of a specific biome type when the other biome type is assigned to the dominant biome. TEGR/COEG for steppe/forest, DRYT/COEG for meadow/forest, COEG/DRYT for forest/meadow, TEGR/DRYT for steppe/meadow, SHRU/DRYT for shrubland/meadow, COEG/TEGR for forest/steppe, DRYT/TEGR for meadow/steppe, DRYT/SHRU for meadow/shrubland, and DESE/TEGR for desert-steppe/steppe. Note that the high meadow scores in

the samples assigned to shrubland and steppe (shown by high ratios of meadow to shrubland or steppe) suggest that shrubland and steppe biomes are meadow-like in most cases, particularly for the last 1.5 Myr. The full names of biome types are presented in Supplementary Table 4. In panel **b**, the units for pollen influx and pollen concentration are 100 grains/cm²/yr and 1000 grains/ml, respectively. Desert-steppe pollen types consist of all the taxa that are assigned in DESE biome scheme.

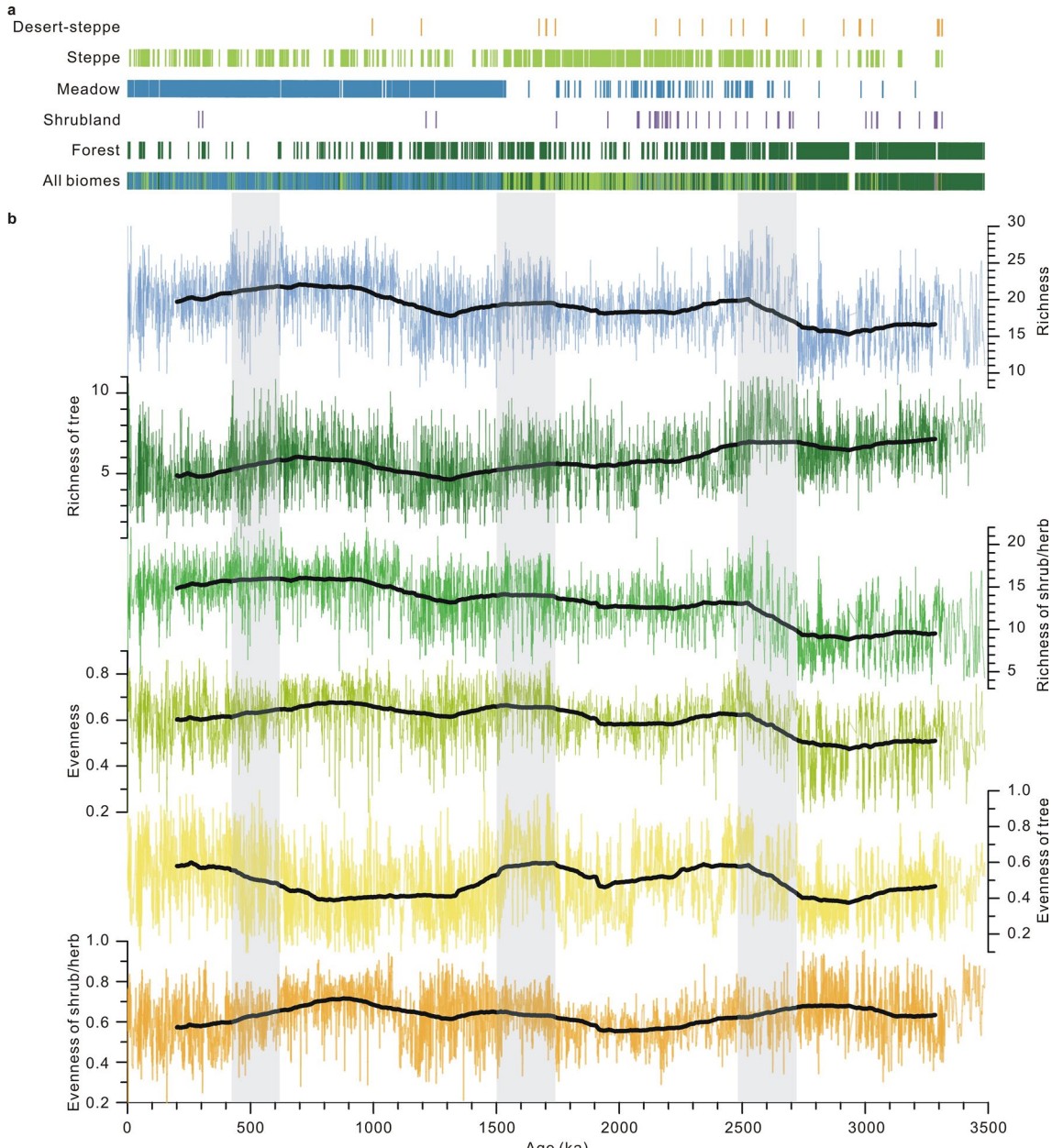

**Extended Data Fig. 4 | Biome and floristic diversity reconstructions from the fossil pollen data of Zoige Basin core. a**, Combined mega-biomes. **b**, Floristic diversity represented by richness and evenness. Panels from top to bottom: rarefied richness of all pollen taxa, tree taxa, and shrub/herb taxa, and evenness of all pollen taxa, tree taxa, and shrub/herb taxa. The bold lines represent 400-kyr running averages of the diversity curves. The gray bars indicate high values for both richness of all pollen taxa and tree taxa, suggesting the temporal edge effects of two ecosystem regimes that are similar to the effects observed on a spatial scale in modern ecological studies.

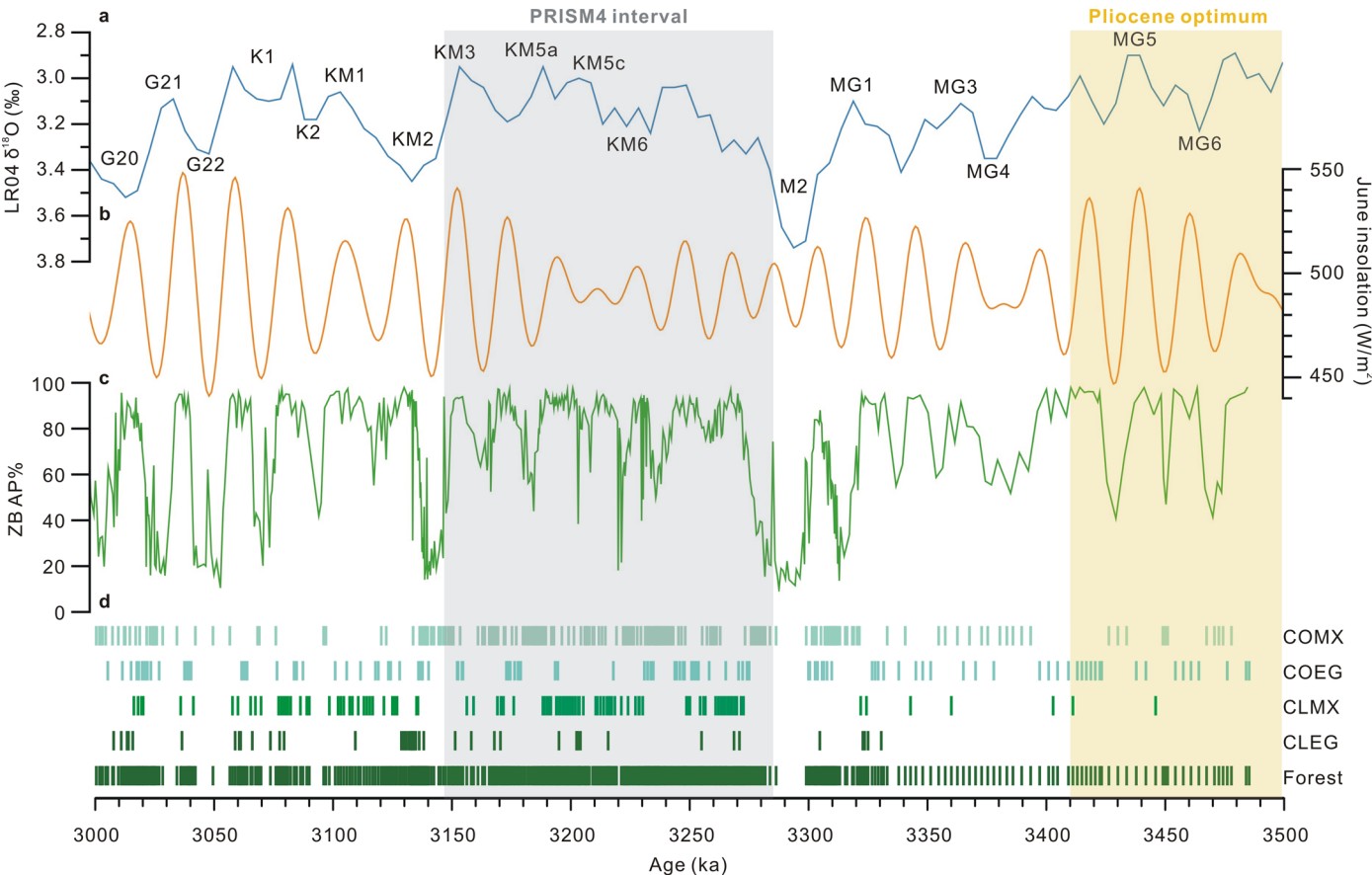

**Extended Data Fig. 5 | Persistent forest cover during the warm mid-Pliocene in the Zoige region. a**, LR04 δ¹⁸O stack[17]. **b**, 30°N June insolation[16]. **c**, AP% from the Zoige Basin. AP% shows strong coherence with changes in summer insolation; however, some deep glacials are clearly imprinted in the record, for example, M2 and KM2. The interglacial KM5c, an astronomical analog of the present, exhibits small variability. **d**, Forest biome. COMX: cool mixed forest; COEG: cool evergreen needle-leaved forest; CLMX: cold-temperate evergreen needle-leaved and mixed forest; CLEG: cold evergreen needle-leaved forest. Forest persists with a few exceptions. Note that the age scale is based on correlation of the proxies with ETP (eccentricity, tilt, reversed precession), yielding a phase difference of several thousand years from that of LR04 stack.

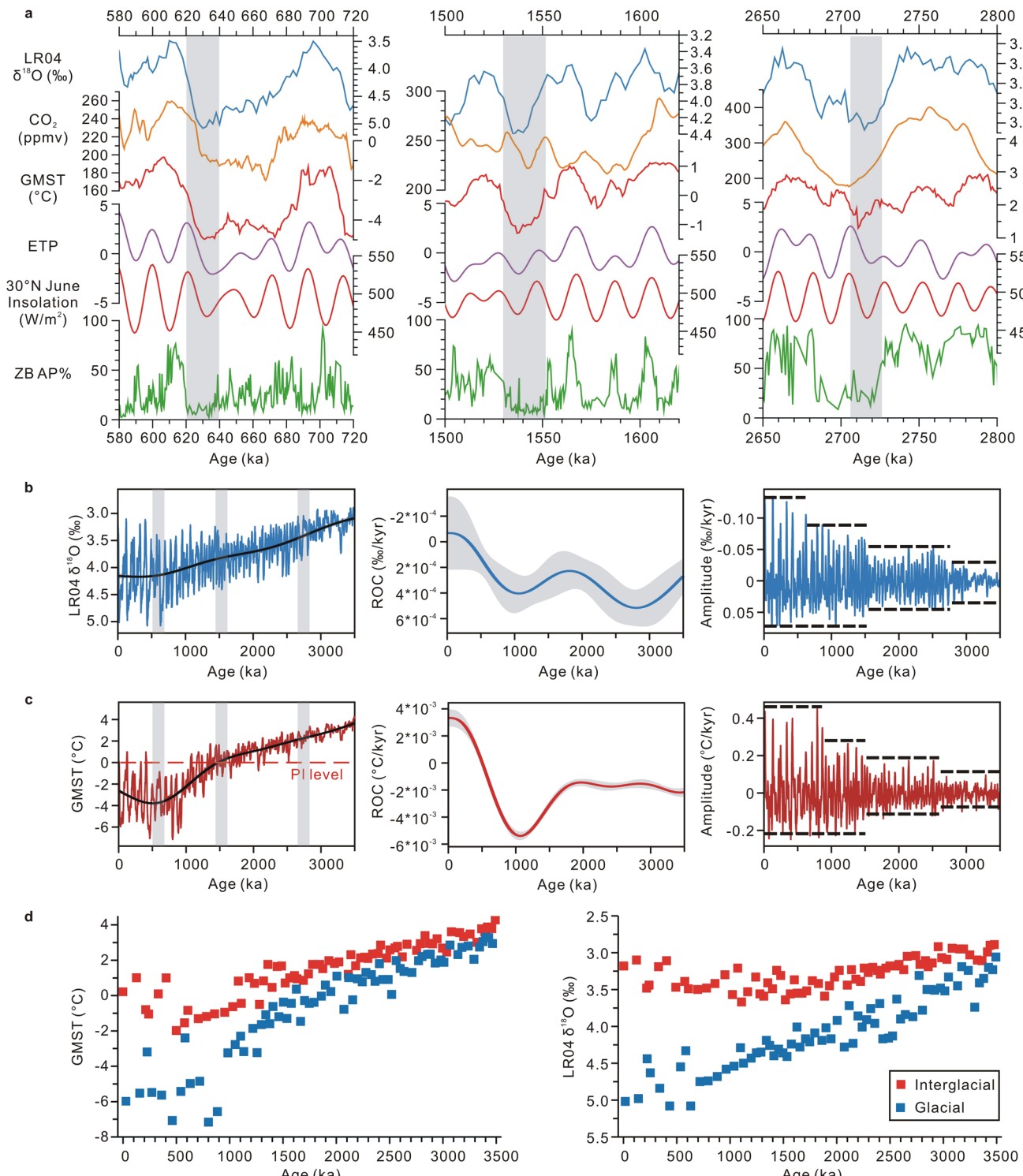

**Extended Data Fig. 6 | Global climate boundaries around the major vegetation regime transformations at ~0.6, ~1.5, and ~2.7 Ma in the Zoige region.** **a**, Correlation of AP% and global climate. Panels from top to bottom: ice volume represented by the LR04 δ[18]O stack[17], $CO_2$ concentration[19], global mean surface temperature (GMST) (°C) relative to pre-industrial (PI) values[20], ETP, June insolation at 30°N[16], and AP%. **b**, **c**, The temporal evolution, rate-of-change (RoC), and variability amplitude of the LR04 δ[18]O stack[17] and GMST relative to PI[20]. Global ice-volume shows a large increase, while GMST shows an obvious

decrease around these three transitions in forest regime (vertical gray bars). The black solid lines are their generalized additive model (GAM) smoothers. Rate-of-change curves are derived from the first order derivative of the GAM smooth lines. The variability amplitude of the LR04 δ[18]O and GMST display stepwise increases as shown by the black dashed lines. **d**, The long-term trends of GMST (left panel) and the LR04 δ[18]O (right panel). The squares represent the maxima and minima for interglacial and glacial intervals.

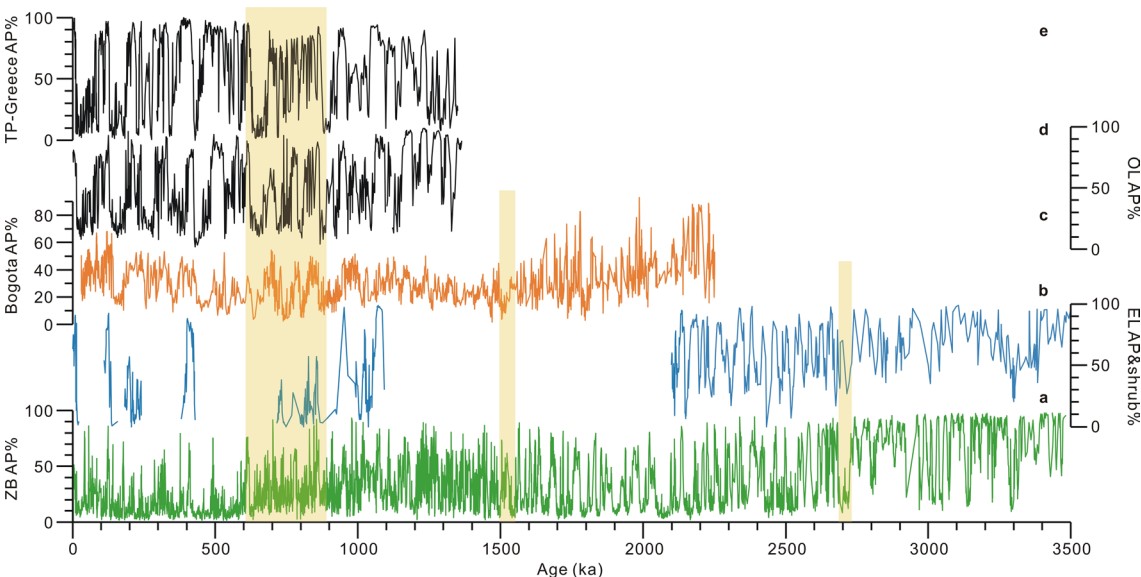

**Extended Data Fig. 7 | Forest decline at ~2.7 Ma, ~1.5 Ma and ~0.9–0.6 Ma revealed by pollen records from various latitudes. a**, Arboreal pollen abundances (AP%) from the Zoige Basin (this study). **b**, AP&Shrub% from Lake El'gygytgyn in the Arctic region[21,22,108]. **c**, AP% excluding *Quercus* and *Alnus* from the Bogotá basin in the tropical high Andes[23], though the decline around 1.5 Ma is also associated with the sedimentary setting change due to rapid subsidence (H. Hooghiemstra, pers. comm.). **d**, AP% from Lake Ohrid, Europe[24]. **e**, AP% from Tenaghi Philippon, Greece (digitized data)[25].

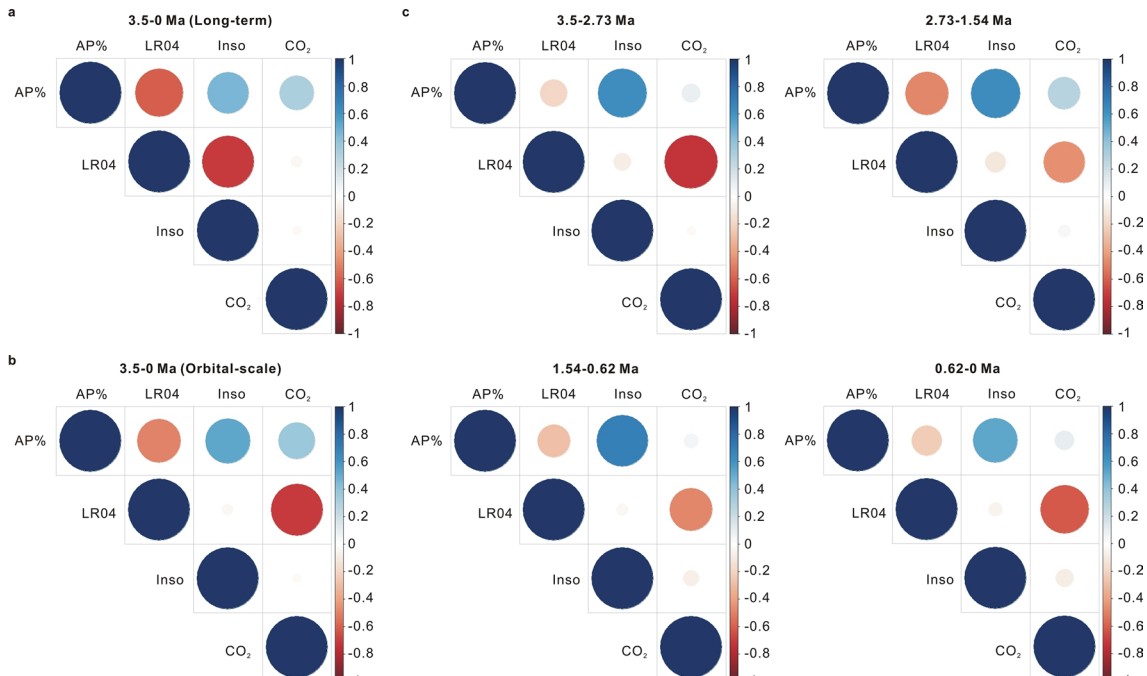

**Extended Data Fig. 8 | Spearman rank correlations between the measured variables in the structural equation model based on the AP% from the Zoige Basin core. a**, Long-term change based on all AP% data indicates that the global ice volume (represented by the LR04 $\delta^{18}O$ stack) is the most important driver.

**b**, **c**, Orbital-scale changes based on band-pass (13–250 kyr) filtered AP% for the past 3.5 Ma and four intervals (3.5–2.73 Ma, 2.73–1.54 Ma, 1.54–0.62 Ma, 0.62–0 Ma) show the dominant control of summer insolation and the strengthened impact of global ice volume.

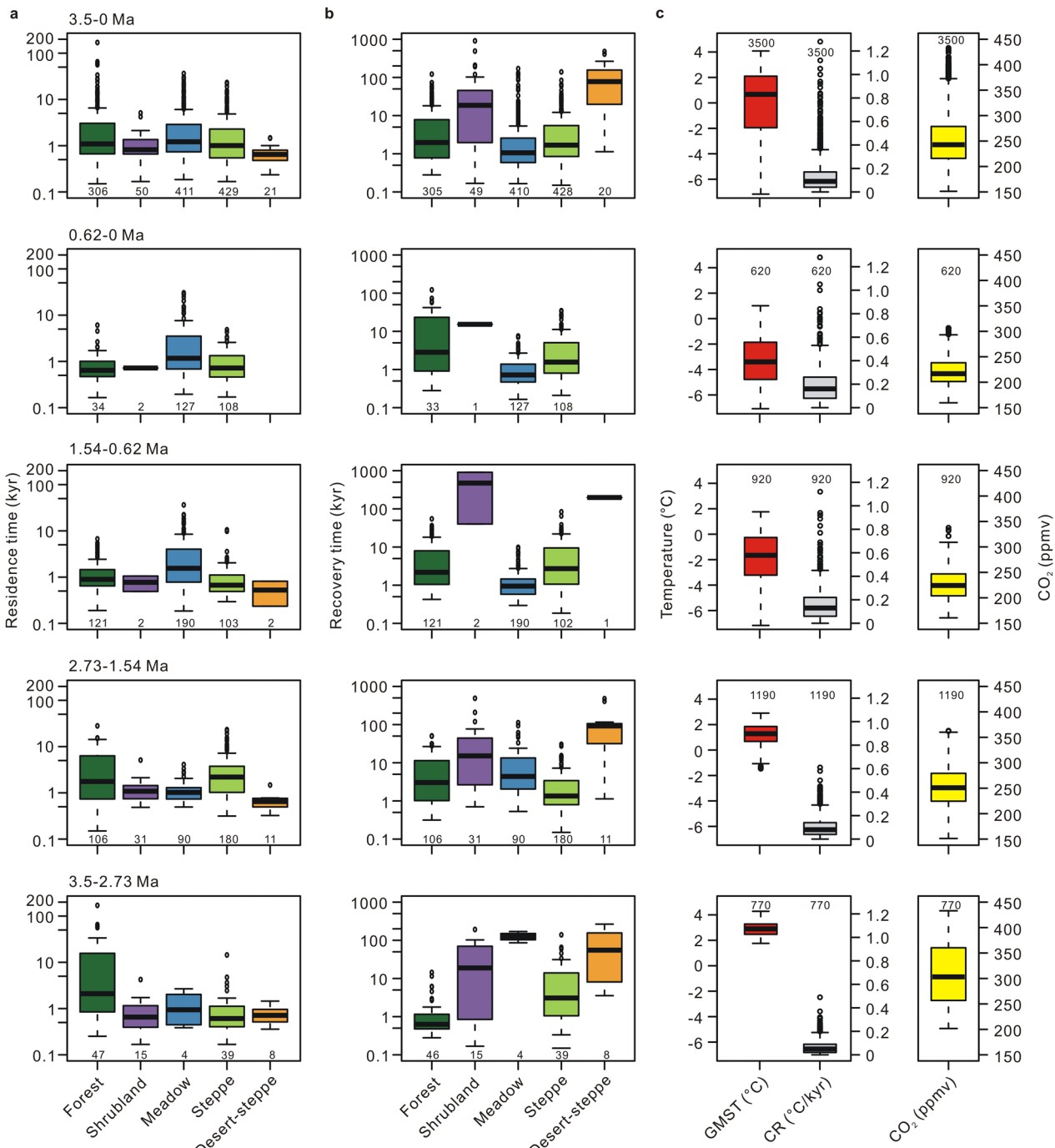

**Extended Data Fig. 9 | Relationships between resilience of the mega-biomes reconstructed from the Zoige Basin core and global climate in corresponding intervals.** Resilience is demonstrated by boxplots of median values and the range of residence time (**a**) and recovery time (**b**), which display medians (central lines), interquartile ranges (boxes), and 25th and 75th centiles plus or minus 1.5 times the interquartile range (whiskers). **c**, Global mean surface temperature (GMST) relative to pre-industrial (PI)[20], GMST change rate (CR), and atmosphere $CO_2$ concentration[19] for the intervals of 3.5–2.73, 2.73–1.54, 1.54–0.62, and 0.62–0 Ma. The sample size used to derive each boxplot (*n*) is listed above or under the boxplot.

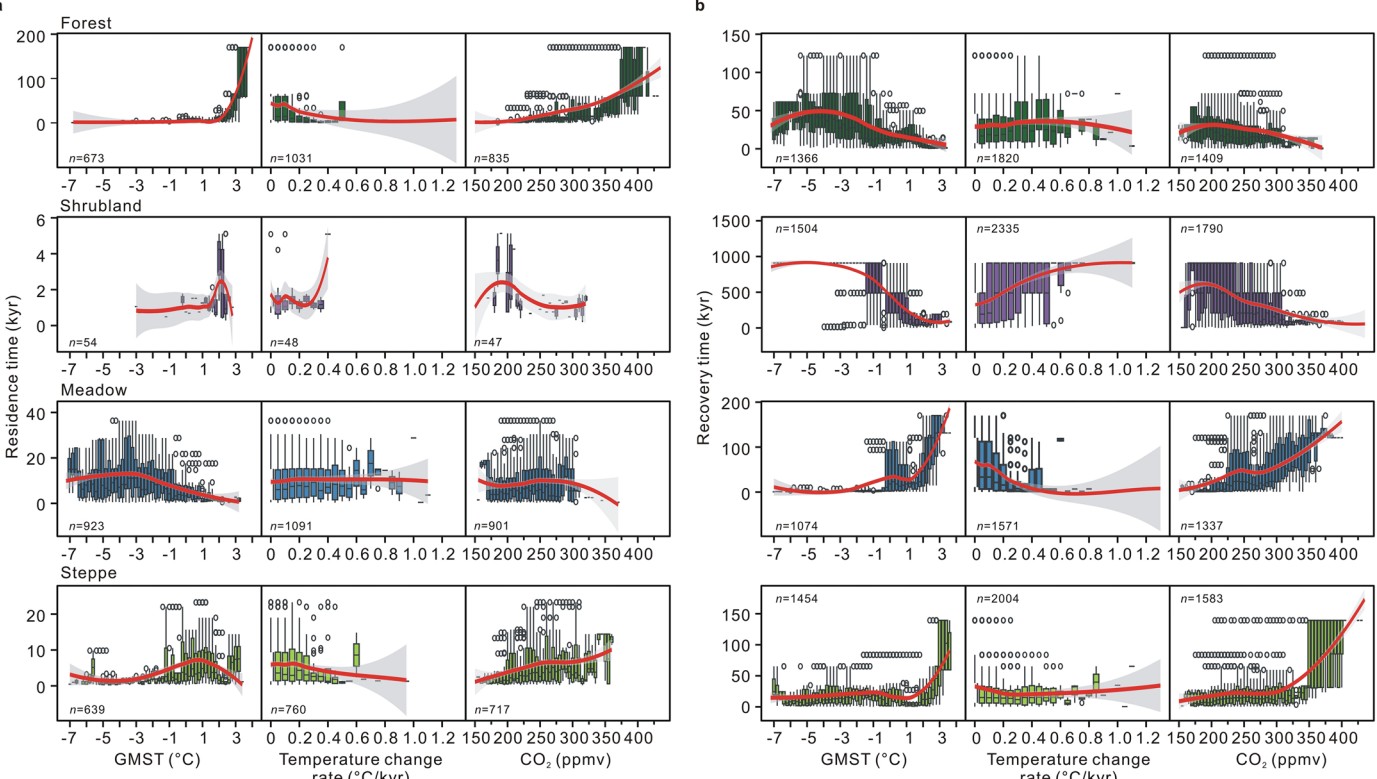

**Extended Data Fig. 10 | Resilience changes of the mega-biomes of Zoige region under different global climate scenarios.** Resilience is demonstrated by the boxplots of median values and the range of residence time (**a**) and recovery time (**b**), which display medians (central lines), interquartile ranges (boxes), and 25th and 75th centiles plus or minus 1.5 times the interquartile range (whiskers). Red line and gray shading in each plot represent the LOESS smoother and its 95% confidence interval. Sample size of each panel (*n*) is defined on the top or bottom left of the panel.

# Reporting Summary

## Statistics

For all statistical analyses, confirm that the following items are present in the figure legend, table legend, main text, or Methods section.

| n/a | Confirmed | |
|---|---|---|
| ☐ | ☒ | The exact sample size (*n*) for each experimental group/condition, given as a discrete number and unit of measurement |
| ☒ | ☐ | A statement on whether measurements were taken from distinct samples or whether the same sample was measured repeatedly |
| ☐ | ☒ | The statistical test(s) used AND whether they are one- or two-sided<br>*Only common tests should be described solely by name; describe more complex techniques in the Methods section.* |
| ☒ | ☐ | A description of all covariates tested |
| ☒ | ☐ | A description of any assumptions or corrections, such as tests of normality and adjustment for multiple comparisons |
| ☐ | ☒ | A full description of the statistical parameters including central tendency (e.g. means) or other basic estimates (e.g. regression coefficient) AND variation (e.g. standard deviation) or associated estimates of uncertainty (e.g. confidence intervals) |
| ☒ | ☐ | For null hypothesis testing, the test statistic (e.g. *F*, *t*, *r*) with confidence intervals, effect sizes, degrees of freedom and *P* value noted<br>*Give P values as exact values whenever suitable.* |
| ☒ | ☐ | For Bayesian analysis, information on the choice of priors and Markov chain Monte Carlo settings |
| ☒ | ☐ | For hierarchical and complex designs, identification of the appropriate level for tests and full reporting of outcomes |
| ☒ | ☐ | Estimates of effect sizes (e.g. Cohen's *d*, Pearson's *r*), indicating how they were calculated |

*Our web collection on statistics for biologists contains articles on many of the points above.*

## Software and code

Policy information about availability of computer code

| Data collection | We obtain the samples from field coring and perform the laboratory analyses by ourselves to collect the data. |
|---|---|
| Data analysis | R 4.3.2, Matlab R2023a, Tilia 1.7.16, LPJ-GUESS 3.1 & 4.1, iLOVECLIM 1.2 and NorESM-L are used for data analyses in this study. |

For manuscripts utilizing custom algorithms or software that are central to the research but not yet described in published literature, software must be made available to editors and reviewers. We strongly encourage code deposition in a community repository (e.g. GitHub). See the Nature Portfolio guidelines for submitting code & software for further information.

## Data

Policy information about availability of data

All manuscripts must include a data availability statement. This statement should provide the following information, where applicable:

- Accession codes, unique identifiers, or web links for publicly available datasets
- A description of any restrictions on data availability
- For clinical datasets or third party data, please ensure that the statement adheres to our policy

The data are present in the paper and/or the Additional information. They are also available in the figshare (https://doi.org/10.6084/m9. figshare.27966036) database.

# Research involving human participants, their data, or biological material

Policy information about studies with human participants or human data. See also policy information about sex, gender (identity/presentation), and sexual orientation and race, ethnicity and racism.

| Reporting on sex and gender | Not applicable. |
|---|---|
| Reporting on race, ethnicity, or other socially relevant groupings | Not applicable. |
| Population characteristics | Not applicable. |
| Recruitment | Not applicable. |
| Ethics oversight | Not applicable. |

Note that full information on the approval of the study protocol must also be provided in the manuscript.

# Field-specific reporting

Please select the one below that is the best fit for your research. If you are not sure, read the appropriate sections before making your selection.

☐ Life sciences    ☐ Behavioural & social sciences    ☒ Ecological, evolutionary & environmental sciences

For a reference copy of the document with all sections, see nature.com/documents/nr-reporting-summary-flat.pdf

# Ecological, evolutionary & environmental sciences study design

All studies must disclose on these points even when the disclosure is negative.

| Study description | Biome changes on the eastern Tibetan Plateau for the past 3.5 million years are reconstructed based on fossil pollen records from the Zoige Basin, and the vegetation resilience and its response to climate changes are discussed. |
|---|---|
| Research sample | Two sediment cores from the Zoige Basin on the eastern Tibetan Plateau. |
| Sampling strategy | An average time resolution of ~700-year are retrieved for the fossil pollen records to capture long-term, orbital and millennial scale vegetation variability for the past 3.5 million years. |
| Data collection | Fossil pollen are extracted from the sediment samples of the drill core following standard procedures, and identified under optical microscope. |
| Timing and spatial scale | Mid-Pliocene to the present (3.5 million years), eastern Tibetan Plateau. |
| Data exclusions | No data are excluded from the analyses. |
| Reproducibility | The experiment and analysis procedures are well described in the manuscript. |
| Randomization | Randomization is not relevant to this study. |
| Blinding | Blinding is not relevant to this study. |

Did the study involve field work?    ☒ Yes    ☐ No

# Field work, collection and transport

| Field conditions | Alpine meadow in the Zoige Basin, eastern Tibetan Plateau. Mean annual temperature and precipitation recorded at the Zoige meteorological station are 1.4°C and 650 mm, respectively. |
|---|---|
| Location | 33°58'03.09"N, 102°20'09.76"E, 3442 m a.s.l. |
| Access & import/export | Field work were permitted by the Forestry and Grassland Bureau of Ruoergai (Zoige) County, and carried out in July-August 2019. All materials are collected in a responsible manner and in accordance with relevant permits and local laws. |
| Disturbance | No disturbance. |

# Reporting for specific materials, systems and methods

We require information from authors about some types of materials, experimental systems and methods used in many studies. Here, indicate whether each material, system or method listed is relevant to your study. If you are not sure if a list item applies to your research, read the appropriate section before selecting a response.

## Materials & experimental systems

| n/a | Involved in the study |
|-----|----------------------|
| ☒ | Antibodies |
| ☒ | Eukaryotic cell lines |
| ☒ | Palaeontology and archaeology |
| ☒ | Animals and other organisms |
| ☒ | Clinical data |
| ☒ | Dual use research of concern |
| ☒ | Plants |

## Methods

| n/a | Involved in the study |
|-----|----------------------|
| ☒ | ChIP-seq |
| ☒ | Flow cytometry |
| ☒ | MRI-based neuroimaging |

## Plants

| | |
|---|---|
| Seed stocks | Not applicable. |
| Novel plant genotypes | Not applicable. |
| Authentication | Not applicable. |

