## [Peer Review File · Nature Ecology & Evolution]

3.5 million years of Tibetan Plateau vegetation dynamics in response to climate change

Corresponding Author: Professor Yan Zhao

Version 0:

Decision Letter:

4th February 2025

Dear Professor Zhao,

Your manuscript entitled "3.5 million years of Tibetan Plateau vegetation dynamics in response to climate change" has now been seen by three reviewers, whose comments are attached. The reviewers have raised a number of concerns which will need to be addressed before we can offer publication in Nature Ecology & Evolution. We will therefore need to see your responses to the criticisms raised and to some editorial concerns, along with a revised manuscript, before we can reach a final decision regarding publication.

We therefore invite you to revise your manuscript taking into account all reviewer and editor comments. Please highlight all changes in the manuscript text file [OPTIONAL: in Microsoft Word format].

* If you have not done so already please begin to revise your manuscript so that it conforms to our Article format instructions at <http://www.nature.com/natecolevol/info/final-submission>. Refer also to any guidelines provided in this letter.

* Extended Data Figures - please ensure that any supplementary figures and tables that are crucial to the manuscript's conclusions are converted into Extended Data figures and tables to increase visibility of these data. Extended Data figures and tables are online-only (present in the online PDF and full-text HTML versions of the paper), peer-reviewed display items that provide essential background to the article but are not included in the main article due to space constraints. A maximum of ten Extended Data display items (figures and tables) is permitted.

Link Redacted

Nature Ecology & Evolution is committed to improving transparency in authorship. As part of our efforts in this direction, we are now requesting that all authors identified as 'corresponding author' on published papers create and link their Open Researcher and Contributor Identifier (ORCID) with their account on the Manuscript Tracking System (MTS), prior to acceptance. ORCID helps the scientific community achieve unambiguous attribution of all scholarly contributions. You can create and link your ORCID from the home page of the MTS by clicking on 'Modify my Springer Nature account'. For more information please visit www.springernature.com/orcid.

[redacted]

Reviewer expertise:

Reviewer #1: global pollen analysis, long term cores

Reviewer #2: Tibetan plateau palynology

Reviewer #3: Asian palaeoclimatology, vegetation reconstruction

Reviewers' comments:

Reviewer #1 (Remarks to the Author):

The manuscript presented by Zhao et al. presents a high temporal resolution record of past environmental change from the Tibetan Plateau for the last 3.5 million years. The high temporal resolution and long duration of this record makes it stand out from most other records of past environmental change globally. The manuscript is well written and organized. The analysis of the data is presented in a clear and organized fashion. Having worked with applying Structural Equation Modelling to palaeoecological data I focused more closely on this aspect of the manuscript and have some suggestions on how this could be further developed. I also have some minor comments on other aspects of the manuscript. See details below.

Structural Equation Model (SEM)

The application of an SEM to this data set is very interesting and I think it offers an important approach to understand (changing) environmental relationships through time. The methods presented here, however, I think could be expanded to make clearer how this approach has been applied. Firstly, it would be very useful to present a visual representation of how the SEM was set up, i.e. what are the links within the models. Secondly, it should be made explicit which proxies are parameterizing which environmental parameter. Thirdly, the question of how the differing chronologies for the various records are aligned needs to be explained; for example, when we ran the SEM model for the Lake Bosumtwi record we iterated the SEM to look at the relationships in multiple different possible chronologies.

Minor comments

Line 211: A lapse-rate is provided here, but it is not clear what this is based on. Please add a reference or explanation of this number.

L304-307: The sentence starting "Grasses are also..." is quite confusing and seems to have some unnecessary 'ands'. Please reword.

L318: Check journal policy on referencing work "in press". Hopefully, this is out soon and can be properly referenced.

L325-326: I think it should be "No significant hiatuses were detected..." also useful to indicate what this detection was based on, i.e. what is it about the sediments that suggests that there are no hiatuses.

L366-384: Pollen analysis section – I am a bit confused about the number of samples examined and the distribution of these through the core. I suggest reorganising the section slightly starting with an indication that a total of 5000 sub-sample were analysed, and then go on to say where they were recovered from. At the moment the sub-sampling strategy seems to jump around a bit. I would add the reference of Stockmarr (1971) in relation to the use of Lycopodium for calculating concentrations. Please provide information on the number of Lycopodium tablets used, the batch number and the Lycopodium concentrations; these data are required for calculating the uncertainty on the concentration quantification. Please also indicate which reference atlases and materials were used to aid identification of the pollen grains. Further, I assume that this extensive data set was compiled by numerous people over many years. It would be nice to record and acknowledge somewhere who counted which sections of the core, and explain how consistency between analysts was assured.

L429-542: Diversity estimation – I think it is useful to calculate the palynological diversity of samples and on a continental

scale I agree that palynological diversity can relate to vegetation diversity, i.e. tropical pollen samples have a higher diversity than temperate pollen samples. However, at the landscape scale large differences in palynological diversity can be caused by factors such as openness of vegetation (e.g. Gosling et al., 2018), and is related to count size (e.g. Keen et al., 2014). It may be interesting to acknowledge this and expand discussion in the context of your site.

L476: The extensive dataset presented (5000 pollen sub-samples) represents an extraordinary amount of work just in terms of the data generation. I would therefore assume that the pollen dataset was generated by multiple researchers over many years. It would be useful to control for differences in analyst, i.e. to demonstrate that key changes in the dataset are not an artefact of analysis by different people.

References

(Note: This list should not be seen a request for citation in you manuscript, but hopefully they are helpful in guiding the suggested developments; other broadly equivalent manuscripts exist within the literature).

Gosling, W.D., Julier, A.C.M., Adu-Bredu, S., Djagbletey, G., Fraser, W.T., Jardine, P.E., Lomax, B.H., Malhi, Y., Manu, E.A., Mayle, F.E. & Moore, S. (2018) Pollen-vegetation richness and diversity relationships in the tropics. *Vegetation History & Archaeobotany* 27, 411–418.

Keen, H.F., Gosling, W.D., Hanke, F., Miller, C.S., Montoya, E., Valencia, B.G. & Williams, J.J. (2014) A statistical sub-sampling tool for extracting vegetation community and diversity information from pollen assemblage data. *Palaeogeography, Palaeoclimatology, Palaeoecology* 408, 48–59.

Stockmarr, J. (1971) Tablets with spores used in absolute pollen analysis. *Pollen et Spore* XIII, 615–621.

William D. Gosling
University of Amsterdam
www.ecologyofthepast.info

Reviewer #2 (Remarks to the Author):

Review of the manuscript “3.5 million years of Tibetan Plateau vegetation dynamics in response to climate change” - by Zhao et al

The manuscript tried to reveal dynamics and evolution of forest, shrubland, meadow, steppe, and desert-steppe over the past 3.5 million years on the Tibetan Plateau using a long and high-resolution pollen record consisting of 5000 pollen samples as well as mature and proved methods. It is a great work of ecology and evolution in Pliocene and Quaternary pollen analysis, and it would also be of interest to the readers of *Nature Ecology & Evolution*. However, the reviewer cannot recommend current manuscript to be published.

Major comments:

1. Although “the Zoige Basin paleo-record highlights that the association of a warmer and stronger monsoon in the Pliocene and early Quaternary outweighed this precipitation effect on meadow when temperature reached certain thresholds” (L269-271), “recent studies indicate that increased precipitation with warming on the TP may promote modern meadow to some degree” (267-269), implying that precipitation and its latent heating have their impacts on the vegetation and landcover of the Tibetan Plateau. Since the manuscript assumes that the dynamics and evolution of vegetation on the Tibetan Plateau over the past 3.5 million years were driven by temperature, however, monsoon dynamics and monsoonal rainfall in this so long time underwent huge changes, precipitation’s role as possible driver on the vegetation dynamics and evolution thus needs more discussions.

2. Traditionally, Chenopodiaceae (as representative plants of desert-steppe) pollen is distinguished from Amaranthaceae pollen. It would be better to explicitly specify its ecological implication and how to use in different methods.

3. Pollen record used in this manuscript is from the Zoige Basin, where sparse vegetation such as desert-steppe might occur during the stadials of glacial periods. In such situations, fossil spectra with very low pollen influx values were dominated by long-distance transported arboreal pollen, leading to the misinterpretation of vegetation types and thus mislocation of ETP tuning points presumably. Although the pollen diagram shows few occurrences of desert-steppe, it is still necessary to rule out this possibility. It is strongly suggested that AP%, total pollen influx, AP pollen influx, desert-steppe pollen%, desert-steppe pollen influx be plotted in pollen diagram such as Figure 3 or new Extended Data Figure (where pollen influx values are estimated by sedimentary rates derived from dating levels and tuning points) to convince readers that reconstructed vegetation types are undoubted.

4. Figure 3:

A, among Pollen percentages of “Trees/shrubs/herbs”, pollen percentages of trees are inconsistent with the sum of pollen percentages of tree taxa critically.

B, the unit of “Pollen sum” may be mislabeled.

C, “Pollen concentration” may miss a unit. Otherwise, they are too low.

Reviewer #3 (Remarks to the Author):

This manuscript reports a rarely continuous 3.5-million-year pollen record from a lake sediment core located on the eastern Tibetan Plateau (TP). By employing multiple methods, the authors reveal remarkable vegetation shifts: from stable forests during the mid-Pliocene, to a coexistence of forest and steppe in the early Quaternary, and ultimately to a meadow-dominated ecosystem since ~1.54 Ma, along with glacial-interglacial and millennial-scale grassland-forest shifts. These shifts are predominantly governed by temperature variations. Based on long-term vegetation trends and global drivers, the authors further identify a critical global warming threshold of ~2–3°C for TP forest expansion and meadow resilience loss. Given the high levels of greenhouse gas emissions, this finding suggests that the TP's meadow ecosystems are currently at risk of substantial transformation.

This work presents a unique and noteworthy conclusion, which I believe will be of immediate interest to the broader scientific community. The whole study is constructed based on comprehensive field campaign, laboratory analysis, and data interpretation. The paper is well-written and structured clearly, and the conclusion is robustly supported by the high-quality figures and accompanying evidence. I recommend publishing the work in *Nature Ecology & Evolution* after minor revisions, please consider the comments below:

1. Line 142: The authors propose that cooling around 2.7 Ma, 1.5 Ma, and 0.9–0.6 Ma may have caused pervasive vegetation transformations for both mountain and low-elevation regions at a global scale. I would recommend incorporating more interpretation of the vegetation transformations (lines 134–142) into figure 4 or adding an additional figure, for better understanding the global signal in Tibetan Plateau as well as other representative sites covering these time spans.
2. Lines 210–213: This is an interesting point as the authors mentioned the possible position change of the alpine treeline in response to ~2–3°C global warming in the mid-Pliocene. For clarity, it would be beneficial if the authors explicitly mentioned the elevation of the Zoige Basin in this context within the text.
3. Lines 264–267: The authors find the elevated temperature would cause the meadow to lose, if global warming exceeds ~1–2°C and ~2–3°C relative to PI. Please discuss more about the critical role of meadows and the consequences of their loss in the stability of the ecosystem in the Tibetan Plateau.
4. Line 265: “transform meadow back to shrubland/steppe and further to forest”. It is possible that forest belts will migrate to higher elevations, replacing meadows and shrubland in a warming climate. The expansion of forest distribution also requires more adequate water conditions. It is suggested that the author explain this appropriately in the text.
5. Please add the drilling site in Figure 1d, to clearly illustrate the modern biome type around the studying site and its relation with the vertical vegetation belts.
6. Figure 2b: The red and green curves are partially interlaced and reduce readability. It is recommended to change one of the colors.
7. To make it easier for the reader to understand and correspond to the text, please label the key vegetation transformation stages in Figure 3 and Figure 4.
8. Please ensure consistency in reference formatting. For instance, refs. 5, 12, and 18 appear to differ from the format of other references.

*****END*****

Version 1:

Decision Letter:

17th April 2025

Dear Dr. Zhao,

Thank you for submitting your revised manuscript "3.5 million years of Tibetan Plateau vegetation dynamics in response to climate change" (NATECOLEVOL-24123435A). It has now been seen again by the original reviewers and their comments are below. The reviewers find that the paper has improved in revision, and therefore we'll be happy in principle to publish it in *Nature Ecology & Evolution*, pending minor revisions to satisfy the reviewers' final requests and to comply with our editorial and formatting guidelines.

If you have not done so already, please ensure that you also email us completed copies of the Reporting summary and Editorial policy checklists:

Reporting summary: https://www.nature.com/documents/nr-reporting-summary.pdf

Editorial policy checklist: https://www.nature.com/documents/nr-editorial-policy-checklist.pdf

We are now performing detailed checks on your paper and will send you a checklist detailing our editorial and formatting

requirements in about a week. Please do not upload the final materials and make any revisions until you receive this additional information from us.

[redacted]

Reviewer #2 (Remarks to the Author):

The authors have emphasized the reviewer's concerns, no more comments and suggestions.

Reviewer #3 (Remarks to the Author):

The authors have thoroughly addressed my previous concerns through expanded datasets, refined analyses, and clearer mechanistic interpretations. This work provides an exceptional perspective to reconstruct past vegetation dynamics with unprecedented and continuous 3.5-million-year pollen record, while offering actionable insights for modern vegetation conservation and future adaptation. I recommend publication in Nature Ecology and Evolution in its current form.

Detailed Responses to the Reviews

We are grateful to the three reviewers for their detailed and insightful comments and valuable suggestions, which have greatly helped us improve our manuscript. Please find below our point-by-point responses (the comments and our responses are in different fonts).

Response to Reviewer #1:

...The high temporal resolution and long duration of this record makes it stand out from most other records of past environmental change globally. The manuscript is well written and organized. The analysis of the data is presented in a clear and organized fashion....The application of an SEM to this data set is very interesting and I think it offers an important approach to understand (changing) environmental relationships through time. The methods presented here, however, I think could be expanded to make clearer how this approach has been applied.

Thanks to Prof. W. Gosling for all the encouragement and the constructive suggestions provided below.

Firstly, it would be very useful to present a visual representation of how the SEM was set up, i.e. what are the links within the models. Secondly, it should be made explicit which proxies are parameterizing which environmental parameter. Thirdly, the question of how the differing chronologies for the various records are aligned needs to be explained; for example, when we ran the SEM model for the Lake Bosumtwi record we iterated the SEM to look at the relationships in multiple different possible chronologies.

We have added a new panel to Figure 5 to provide a visual representation of the relationships within the model, to highlight the hypothesized links examined in this study. The diagram specifies which proxies were used to parameterize particular environmental parameters, e.g., global ice volume is represented by marine benthic $\delta^{18}\text{O}$. Since our SEM analysis focused solely on global drivers (ice volume, low-latitude summer insolation, and global atmospheric CO_2), we did not include regional environmental parameters in our paper. We are currently working on generating additional multiproxy data from our core, such

as fire and precipitation proxies. This will enable us to explore the relationship between vegetation and the regional environment in greater depth in the future, building on the excellent research conducted by Gosling et al. (2022). In the context of this study, the conceptual model is based on our understanding of the global factors influencing vegetation dynamics in the monsoonal region of the eastern Tibetan Plateau, as supported by previous studies (Shen et al., 2005; Zhao et al., 2020).

In response to the Reviewer’s comment, we have also expanded the Methods section to provide additional details and included a new Supplementary Table 5. This table presents the standardized parameter estimates, model performance metrics, and the intercorrelational relationships among the predictor variables.

Hypothesized relationships examined in this study. Measured variables are indicated by square gray boxes

The age model of the Zoige record was constructed by aligning tree pollen abundance (AP%) with an ETP record (generated by normalizing and averaging eccentricity, tilt, and reversed precession) and with the LR04 benthic δ¹⁸O stack, based on an initial independent chronological framework derived from paleomagnetic data, AMS radiocarbon dating, and OSL dating. To avoid circular reasoning, the final age model was established using ETP correlation control points, which were critical for exploring the influence of high-latitude ice volume forcing on vegetation changes. Given that our approach differs from

that used for Lake Bosumtwi (Gosling et al., 2022), we did not employ Bacon to generate multiple alternative versions of the chronology.

We recognize that aligning the differing chronologies across various records will introduce some uncertainties. The age differences between the LR04 stack and simulated CO₂ records (Lisiecki & Raymo, 2005; Stap et al., 2016) are negligible, whereas they differ by several thousand years when compared to summer insolation (Laskar et al., 2004). Additionally, the Zoige AP% record, with its age tuned to the ETP, exhibits a phase difference of up to several thousand years when compared to the LR04 stack and global atmospheric CO₂ records, even though the ETP also incorporates obliquity and eccentricity. To address these discrepancies, we performed additional SEM analyses using an alternative chronology obtained by tuning AP% to the LR04 δ¹⁸O stack record. The results, as illustrated in the figure below, consistently underscore the fundamental role of insolation on the orbital scale, the growing influence of global ice volume on long-term change, and the secondary role of CO₂, thereby reinforcing the robustness of our earlier conclusions.

Panel A: Long-term change (all data are used for analysis); **Panel B:** Orbital change (bandpass filtered data of 250-13 ka are used for analysis). The age of AP% are based on the tuning to the LR04 stack

Given the specific research questions we aim to address and the 3.5-million-year time span of the study, our alignment methodology and the overarching conclusions regarding the dynamic global drivers remain robust, despite potential uncertainties arising from age discrepancies among the parameters. In any case, we have discussed this issue and the associated uncertainties in the Methods section.

Minor comments

Line 211: A lapse-rate is provided here, but it is not clear what this is based on. Please add a reference or explanation of this number.

We have rewritten this sentence and added relevant references: “During the mid-Pliocene, global warming of ~2–3°C would have elevated the upper tree-line by several hundred meters, assuming a temperature lapse rate of ~0.62–0.65°C per 100 m, based on commonly used values and regional meteorological observations (Immerzeel et al., 2012; Guo et al., 2016).”

Reference:

Immerzeel, W.W. et al., Hydrological response to climate change in a glacierized catchment in the Himalayas. *Clim. Change* 110(3–4), 721–736 (2012).

Guo, X.Y. et al., Spatio-temporal variability of vertical gradients of major meteorological observations around the Tibetan Plateau. *Int. J. Climatol.* 36, 1901–1916 (2016).

L304-307: The sentence starting “Grasses are also...” is quite confusing and seems to have some unnecessary ‘ands’. Please reword.

We have reworded this sentence: “Grasses are also important components in these two vegetation belts, primarily including *Elymus nutans*, *E. burchanbuddae*, *Stipa capillacea*, *S. aliena*, and *S. purpurea*.”

L318: Check journal policy on referencing work “in press”. Hopefully, this is out soon and can be properly referenced.

This paper has recently been published online (Ren et al., Land 2024). We have updated this citation and included it in the reference list.

L325-326: I think it should be “No significant hiatuses were detected...” also useful to indicate what this detection was based on, i.e. what is it about the sediments that suggests that there are no hiatuses.

We have changed it to “No significant sampling gaps were detected except a hiatus from 2964 ka to 2934 ka”, considering this paragraph focuses solely on core drilling.

In addition to the high drilling recovery, we have provided further evidence supporting the near-continuous nature of the sediments in the paragraph about lithology in this section. First, the lithology is predominantly composed of lacustrine-originated clay, with the exception of two fluvial sandy layers at the top. Second, the sediments exhibit clear stratification without apparent large discontinuities. Third, the tuning of AP% with ETP between the paleomagnetic reversal levels is quite straightforward (Fig. 2 in the main text, also shown below), indicating a generally stable sedimentation rate. In response to the suggestion, we have added a short paragraph in this section to clarify this.

Left panel: Image of typical clay in the core; **Right panel:** Correlation (marked by dotted lines) of AP% with ETP (Laskar et al., 2004). Solid orange lines: paleomagnetic reversal levels.

L366-384: Pollen analysis section

I am a bit confused about the number of samples examined and the distribution of these through the core. I suggest reorganising the section slightly starting with an indication that a total of 5000 sub-sample were analysed, and then go on to say where they were recovered from. At the moment the sub-sampling strategy seems to jump around a bit.

We have reorganized this section as follows: A total of 5,000 samples were analyzed from the combined cores ZB13-C2 and ZB19-C1. For the new core ZB19-C1, 2,088 pollen subsamples of ~2 cm³ in volume were collected at intervals of ~40 cm, except for the Pliocene section, where samples were taken at 20-10 cm intervals to capture detailed pollen and vegetation changes during this critical period for the data-model comparison project. For core ZB13-C2, a total of 2,912 samples were analyzed, including 2,787 previously published samples and 125 newly analyzed samples to achieve higher resolution data for certain interglacial periods.

*I would add the reference of Stockmarr (1971) in relation to the use of *Lycopodium* for calculating concentrations. Please provide information on the number of *Lycopodium* tablets used, the batch number and the *Lycopodium* concentrations; these data are required for calculating the uncertainty on the concentration quantification.*

In our study, a tablet containing a known number of *Lycopodium* spores (Lund batch #1031; 20,848 grains/tablet) was added to each sample at the beginning of the process for calculation of pollen concentration.

We have added the reference of Stockmarr (1971) and the information about the *Lycopodium* tablets in the revised Methods.

Please also indicate which reference atlases and materials were used to aid identification of the pollen grains.

Pollen identifications were conducted using published literature on Chinese pollen morphology and reference collections at the Institute of Geographic Sciences and Natural Resources Research, Chinese Academy of Sciences (IGGNRR, CAS). We have specified the reference atlases and materials used.

References:

Wang, F. X., *et al.* Pollen Flora of China (2nd Edition) (Science Press, Beijing, 1995).

Xi, Y. Z. & Ning, J. C. Pollen morphology of plants from Chinese arid and semiarid areas. *Yushania* 11, 119–191 (1994).

Tang, L. Y., *et al.* An Illustrated Handbook of Quaternary Pollen and Spores in China (Science Press, Beijing, 2016).

Further, I assume that this extensive data set was compiled by numerous people over many years. It would be nice to record and acknowledge somewhere who counted which sections of the core, and explain how consistency between analysts was assured.

We sincerely appreciate the reviewer's recognition of our laborious efforts in generating the pollen data. Over the past 11 years, ten researchers in our group have contributed to the pollen counting process, with eight individuals involved in core ZB13-C2 and six in core ZB19-C1, four of whom worked on both cores. In the initial submission, we acknowledged all pollen analysts in the "Author Contributions" section. Following the reviewer's good suggestion, we have further clarified the specific sections of each core analyzed by each individual in the figshare dataset.

Regarding consistency between analysts, we made significant efforts to minimize uncertainties throughout the entire process, from laboratory strategy design, sample preparation to pollen counting. As four out of the ten researchers involved in both cores are experienced in Tibetan Plateau pollen analysis, all samples were pretreated in the same laboratory, and identifications were conducted in the same microscope room, they helped to facilitate coordination for the long pollen sequence.

The major measures taken include the following:

i) Prior to initiating pollen analysis, we dedicated nearly three months to make preparations. This mainly involved standardizing the pretreatment protocol, creating reference pollen slides based on our previous studies in the region and taking the images of major pollen taxa from samples at different levels of the core, consulting experts to identify challenging pollen taxa, and establishing

uniform counting rules. ii) In addition to ongoing communication among analysts, we organized regular meetings (weekly during the initial phase) to address issues related to pollen identification and counting rules. iii) To verify consistency between analysts, we conducted multiple rounds of cross-checking. This included having one analyst review the entire core and exchanging selected samples from different sections between analysts for pollen counting. Encouragingly, these cross-checks demonstrated high consistency in results for the same sample, with differences falling within an acceptable range (e.g., majority of AP% difference range being <3%), as illustrated in the diagram below.

We have added one paragraph in the Methods to briefly explain how we assured analytical consistency.

L429-542: Diversity estimation – I think it is useful to calculate the palynological diversity of samples and on a continental scale I agree that palynological diversity can relate to vegetation diversity, i.e. tropical pollen samples have a higher diversity than temperate pollen samples. However, at the landscape scale large differences in palynological diversity can be caused by factors such as openness of vegetation (e.g. Gosling et al., 2018), and is related to count size (e.g. Keen et al., 2014). It may be interesting to acknowledge this and expand discussion in the context of your site.

We agree with the suggestion. In the light of the reviewer's constructive comment, we have added more discussions in revised Methods.

Indeed, the representation of palynological diversity (PD) in relation to vegetation diversity can be influenced by multiple factors, such as vegetation openness (Gosling et al., 2018), count size (Keen et al., 2014), and the taxonomic precision of pollen identification (Birks et al., 2016). However, palynological diversity can remain comparable within a single core over an extended time series (Xiao et al., 2008; Birks et al., 2016; Liang et al., 2019), provided that count size, taxonomic precision, and laboratory protocols remain consistent over time. Among these factors, vegetation openness may introduce greater uncertainties in long time series compared to others. In our study region, the Zoige Basin, previous research on Holocene PD has shown that vegetation openness plays a secondary role (Liang et al., 2019). Nevertheless, to account for the potential influence of openness on total diversity, we also separately analyzed the diversities of tree taxa and shrub/herb taxa.

Given the extensive dataset we have generated, this paper for *Nature Ecology & Evolution* focuses primarily on vegetation transformation in response to climate, adhering to the journal's word limit. A more detailed discussion on palynological diversity (PD) and dynamics will be presented in a separate manuscript. We greatly appreciate your suggestion and will incorporate it to enhance our data interpretation.

L476: The extensive dataset presented (5000 pollen sub-samples) represents an extraordinary amount of work just in terms of the data generation. I would therefore assume that the pollen dataset was generated by multiple researchers over many years. It would be useful to control for differences in analyst, i.e. to demonstrate that key changes in the dataset are not an artefact of analysis by different people.

In our response to the comment on "L366-384: Pollen analysis section," we have detailed the measures we took to coordinate and minimize differences among analysts. Based on careful investigation, we concluded that the key changes in the dataset reflect real features rather than artefacts of analytical differences.

First, cross-checks revealed that the differences among analysts fall within an acceptable range. Second, the boundaries of the key changes at ~2.73 Ma, 1.54 Ma, and 0.62 Ma do not coincide with transitions between different analysts. Third, arboreal pollen percentages (AP%) in core ZB13-C2 clearly show consistent changes with Rb/Sr and carbonate percentages (Carb%) (as shown in the figure below) in terms of key change shifts, glacial-interglacial variations, and even multi-millennial-scale changes (Zhao et al., 2020).

Results from multiproxy analysis of core ZB13-C2. Black line denotes 9-point running mean of Rb/Sr. In the revised Methods section, we have added more details about the controls implemented to address differences among analysts (refer to our response above) and have explained why the key changes are not artefacts of differing analysts.

References

We have cited all three references (Gosling, W.D. et al., 2018; Keen et al, 2014; Stockmarr 1971) as suggested, which are helpful for discussing diversity and pollen concentration.

Response to Reviewer #2:

It is a great work of ecology and evolution in Pliocene and Quaternary pollen analysis, and it would also be of interest to the readers of Nature Ecology & Evolution.

We thank the reviewer for all the encouragement and the valuable suggestions provided below

Major comments:

1. Although “the Zoige Basin paleo-record highlights that the association of a warmer and stronger monsoon in the Pliocene and early Quaternary outweighed this precipitation effect on meadow when temperature reached certain thresholds” (L269-271), “recent studies indicate that increased precipitation with warming on the TP may promote modern meadow to some degree” (267-269), implying that precipitation and its latent heating have their impacts on the vegetation and landcover of the Tibetan Plateau. Since the manuscript assumes that the dynamics and evolution of vegetation on the Tibetan Plateau over the past 3.5 million years were driven by temperature, however, monsoon dynamics and monsoonal rainfall in this so long time underwent huge changes, precipitation’s role as possible driver on the vegetation dynamics and evolution thus needs more discussions.

The vegetation in the eastern Tibetan Plateau is strongly influenced by the Asian summer monsoon. A stronger monsoon, characterized by warmer and wetter conditions, generally promotes the expansion of tree populations. However, in the alpine Zoige region, vegetation changes within the meadow-conifer forest ecotone are primarily regulated by temperature (Shen et al., 2005; Zhao et al., 2011, 2020, 2021), as moisture availability is generally sufficient due to the combination of low temperatures and moderate precipitation levels. This is supported by several lines of evidence: the distinct elevational distribution of modern vegetation (Fig. 1); the predominance of forest, meadow, and meadowic steppe biomes over the past 3.5 million years (Extended Data Fig. 3); the strong correlation between arboreal pollen percentages (AP%) and the first axis of a principal component analysis (PCA) of the pollen data (Extended Fig. 2e), with the close relationship between temperature and PCA axis 1, as well as the lack of significance of precipitation in the significance tests of quantitative reconstructions for core ZB13-C2 (Zhao et al., 2020, 2021).

However, this explanation of the overall temperature driver does not necessarily exclude the influence of precipitation. We agree with the reviewer

that monsoon dynamics and monsoonal rainfall have undergone significant changes since the mid-Pliocene, characterized by long-term declines and glacial-interglacial variability. Our previous study highlighted the coupled changes in precipitation and temperature in the Zoige region over the past ~1.7 million years on both long-term and orbital scales (Zhao et al., 2020). On one hand, precipitation can further enhance forest growth, complementing the primary role of temperature. On the other hand, drought stress during periods of weak monsoon intensity or glacial intervals would also limit forest growth and alter grassland composition. This is particularly evident during certain stadials of cold glacial periods, as indicated by the relatively high percentages of Amaranthaceae.

In the original manuscript, we briefly addressed the role of precipitation in several sections (e.g., L146-148, L213-216, L227-231, L436-438). In response to the reviewer's suggestion, we have now further elaborated on this topic (P6, L213-228). They include a discussion of the regional climate drivers influencing vegetation dynamics, with a more explicit emphasis on the impact of precipitation, particularly during periods of weak monsoon intensity or glacial intervals. The reviewer's comments also encourage us to pursue further transient vegetation simulations in future studies.

2. Traditionally, Chenopodiaceae (as representative plants of desert-steppe) pollen is distinguished from Amaranthaceae pollen. It would be better to explicitly specify its ecological implication and how to use in different methods.

As indicated by the reviewer, the families Chenopodiaceae and Amaranthaceae were traditionally treated as separate taxonomic units under classical taxonomy systems. However, the modern phylogeny system APG IV (The Angiosperm Phylogeny Group, 2016) has merged both families into a single taxon – the family Amaranthaceae.

We have adopted this recently widely accepted nomenclature for flowering plant families. In fact, nearly all Amaranthaceae pollen in our study belongs to its subfamily Chenopodioideae (Zhao et al., 2020), which is characteristic of highly continental climate with cold winter and dry summers. We have provided

further details in the Methods section regarding the family name and its ecological implications (P11, L437-441).

3. Pollen record used in this manuscript is from the Zoige Basin, where sparse vegetation such as desert-steppe might occur during the stadials of glacial periods. In such situations, fossil spectra with very low pollen influx values were dominated by long-distance transported arboreal pollen, leading to the misinterpretation of vegetation types and thus mislocation of ETP tuning points presumably. Although the pollen diagram shows few occurrences of desert-steppe, it is still necessary to rule out this possibility. It is strongly suggested that AP%, total pollen influx, AP pollen influx, desert-steppe pollen%, desert-steppe pollen influx be plotted in pollen diagram such as Figure 3 or new Extended Data Figure (where pollen influx values are estimated by sedimentary rates derived from dating levels and tuning points) to convince readers that reconstructed vegetation types are undoubted.

This is a very insightful comment. We thank the reviewer for providing us with the opportunity to further examine the pollen data. Following the reviewer's suggestion, we have incorporated a new panel to Extended Data Figure 3 (considering the figure number limit required by the journal) to display the following data: total pollen concentration, total pollen influx, AP%, AP pollen influx, desert-steppe pollen%, desert-steppe pollen influx, as well as the pollen influx of *Artemisia* and *Amaranthaceae*.

Indeed, long-distance transported arboreal pollen could potentially distort the reconstruction of vegetation types in conditions of sparse vegetation under cold/dry climates. However, this is not likely the case for our record. The following observations support the reliability of our vegetation type reconstruction and the robustness of the ETP tuning.

First, the AP% shows a clear positive relationship with total pollen influx. For clear demonstration reason, we here develop a discussion by taking the samples classified as belonging to the forest and desert-steppe biomes as examples as illustrated in the figure below. High AP% values do not appear to coincide with extremely low pollen concentration and low pollen influx in both

cases. In particular, the samples assigned to the desert-steppe biome do not show very low pollen concentrations or influx.

A. Pollen influx diagram for samples classified as belonging to the forest biome

B. Pollen influx diagram for samples classified as belonging to the steppe-forest biome

Second, the vegetation composition and structure for the samples assigned to the desert-steppe biome (n=23) provide further evidence for this issue. The desert-steppe biome assignment was primarily due to their relatively high abundance of Amaranthaceae and *Artemisia* (ranging from ~10–30%, with only

one sample reaching ~40%), despite tree pollen percentages being around 20–40%. It is also noteworthy that the biomization approach considers plant functional types as a whole.

Our previous synthesis study revealed that in typical desert-steppe regions of China with sparse vegetation, *Artemisia* and Amaranthaceae constitute approximately $\sim 70\% \pm 25\%$ of the total terrestrial pollen percentages (Zhao et al., 2012), which is significantly higher than the values observed in our cases. The distinct vegetation composition and structure in the Zoige Basin region imply that vegetation cover in our study area was denser and less susceptible to issues related to extra-source tree pollen, even during the stadials of glacial periods. Given that the sparse vegetation is currently found at elevations above 4400 m (Shen et al., 2005), where temperatures are $>\sim 6.5^\circ\text{C}$ lower than at our study site due to the ~ 1000 m elevation difference, this assumption is reasonable.

Third, the coherent changes in AP% with the XRF-based Rb/Sr ratio and carbonate content (Carb%) from core ZB13-C2 across various timescales (from supra-orbital to orbital and sub-orbital variabilities) (Zhao et al., 2020; please also refer to the figure in the response to Reviewer #1) further confirm that the AP% exaggeration problem due to locally sparse vegetation likely has a very limited impact.

Fourth, in terms of tuning AP% to ETP, we primarily relied on the orbital-scale variations in AP% rather than a single AP% value. Additionally, the occasional occurrences of desert-steppe vegetation are intermittent, excluding potentials of misplacing tuning points. We have further checked our data and verified that all the AP% peaks used for tuning correspond to high pollen influx, and none of them are from desert-steppe samples. In the ZB13-C2 tuning, the alignment of these points is also supported by Rb/Sr ratios and Carb% (Zhao et al., 2020).

In summary, we believe our reconstructions of vegetation types and orbital tuning are fundamentally not distorted by potential extra-source arboreal pollen. While there may be minor impacts in occasional cases, these do not affect any of the conclusions reached in our paper. We have included additional

discussion to further emphasize the robustness of the biomization results (P13, L515-531; Methods).

4. *Figure 3:*

A, among Pollen percentages of “Trees/shrubs/herbs”, pollen percentages of trees are inconsistent with the sum of pollen percentages of tree taxa critically.

We appreciate the reviewer’s careful observation. All data analysis and diagram plotting were conducted using Tilia 1.7.16 software. We have thoroughly reviewed our plotting procedure. The seemingly inconsistency between the pollen percentages of trees in the “trees/shrubs/herbs” stack plot and the sum of pollen percentages of individual tree taxa is due to the display limitations of the stack plot in the software. This issue arises from the large sample size, which can cause overlapping and conceal some information.

To address this problem, we have revised the diagram by plotting the pollen percentages of trees, trees & shrubs, and trees & shrubs & herbs separately in the Tilia software. These components were then manually stacked in CorelDraw software. This approach ensures a more accurate representation of the data.

B, the unit of “Pollen sum” may be mislabeled.

We have relabeled the unit as “grains” at the top of the “Pollen sum” column. Please also refer to our response below for further details.

C, “Pollen concentration” may miss a unit. Otherwise, they are too low.

In the originally submitted manuscript, we displayed the unit (*100 grains/ml) for “Pollen concentration” at the bottom of Figure 3, but it was mislabeled to the left of this column. We have corrected this by moving the unit to the top of the column.

Response to Reviewer #3:

This work presents a unique and noteworthy conclusion, which I believe will be of immediate interest to the broader scientific community. The whole study is constructed

based on comprehensive field campaign, laboratory analysis, and data interpretation. The paper is well-written and structured clearly, and the conclusion is robustly supported by the high-quality figures and accompanying evidence.

We are grateful for these positive words!

1.Line 142: The authors propose that cooling around 2.7 Ma, 1.5 Ma, and 0.9–0.6 Ma may have caused pervasive vegetation transformations for both mountain and low-elevation regions at a global scale. I would recommend incorporating more interpretation of the vegetation transformations (lines 134-142) into figure 4 or adding an additional figure, for better understanding the global signal in Tibetan Plateau as well as other representative sites covering these time spans.

We have incorporated a new figure (Extended Data Figure 7) to present the correlation with four high-resolution pollen records, although none of them cover the entire time span of the Zoige Basin record. Nevertheless, the correlations indicate that the vegetation transformations occurring at ~2.7 Ma, ~1.5 Ma, and 0.9-0.6 Ma may be globally pervasive, as discussed in the main text.

Forest decline at ~2.7 Ma, 1.5 Ma and ~0.9-0.6 Ma as revealed by pollen records across various latitudes. (a) Arboreal pollen abundances (AP%) from Zoige Basin (ZB; this study); **(b)** AP&Shrub% from Lake El'gygytgyn (EL), Arctic region (Melles et al., 2012; Brigham-Grette et al., 2013; Zhao et al., 2018); **(c)** AP% excluding *Quercus* and *Alnus* from Bogotá basin, tropical high Andes (Torres et al., 2013); **(d)**

AP% from Lake Ohrid, Europe (OL) (Donders et al., 2021); (e) AP% from Tenaghi Philippon (TP), Greece (Tzedakis et al., 2006; digitized data).

2. Lines 210-213: This is an interesting point as the authors mentioned the possible position change of the alpine treeline in response to ~2–3°C global warming in the mid-Pliocene. For clarity, it would be beneficial if the authors explicitly mentioned the elevation of the Zoige Basin in this context within the text.

The Zoige Basin is situated at an elevation of approximately 3350–3450 meters above sea level (a.s.l.). Given the current upper elevational limit of conifer forests, which ranges between ~3800 and ~4000 m, the Zoige Basin is highly sensitive to treeline migration. This elevational context provides a valuable framework for discussing the dynamics of alpine treeline changes.

In the original submitted manuscript, we provided a detailed description of the regional settings (including elevation) of the Zoige Basin in the Methods section. In line with this comment and another related comment below regarding modern vertical vegetation belts, we have added a new subsection titled “Regional Setting, drilling, and core analyses” to the main text to improve the manuscript's structure and clarity.

3. Lines 264-267: The authors find the elevated temperature would cause the meadow to lose, if global warming exceeds ~1–2°C and ~2–3°C relative to PI. Please discuss more about the critical role of meadows and the consequences of their loss in the stability of the ecosystem in the Tibetan Plateau.

Meadows constitute more than ~60% of the alpine grassland area on the Tibetan Plateau (TP) (Wang et al., 2022, 2023). They play a vital role in water conservation, soil preservation, carbon sequestration, biodiversity protection, climate regulation, and the provision of ecological services at both regional and global scales. Consequently, the degradation of alpine meadows on the TP could threaten ecosystem stability through a series of biological, biochemical, and biophysical processes.

We have expanded the related discussions and included additional reference in the "Future Implications" subsection.

4. Line 265: "transform meadow back to shrubland/steppe and further to forest". It is possible that forest belts will migrate to higher elevations, replacing meadows and shrubland in a warming climate. The expansion of forest distribution also requires more adequate water conditions. It is suggested that the author explain this appropriately in the text.

This is a very good point. Coherent variations in AP%, Rb/Sr and Carb% of Zoige (as displayed in the response to Reviewer#1) suggest coupled changes of precipitation and temperature, as evidenced in many studies for monsoon regions. That is, during past warming periods, precipitation also increased at higher elevations. Our fossil pollen records indicate that precipitation levels were sufficient to support the migration of trees to higher elevations during past warm periods, such as the mid-Pliocene, when global temperatures were approximately 3°C higher than pre-industrial (PI) levels. Therefore, we infer that, with projected increases in precipitation alongside global warming, moisture availability would be sufficient to facilitate tree migration to higher elevations. However, we acknowledge that, while temperature plays a predominant role in determining the upper treeline, the rate of treeline migration would likely be also constrained by moisture stress, as evidenced by numerous observations and a recent modeling study (Xu et al., 2025).

In response to this comment, we have clarified the potential role of moisture in treeline upward migration and cited new a reference in the main text (P7, L255-257), which we think is a more appropriate section to address this issue.

5. Please add the drilling site in Figure 1d, to clearly illustrate the modern biome type around the studying site and its relation with the vertical vegetation belts.

We have marked the elevation of the drilling site (~3440 m) in Figure 1d. The site is currently situated within the subalpine meadow belt, while the surrounding mountains, ranging from ~3000 to 3800 m, fall within the subalpine dark coniferous forest belt. In the revised main text, a description of the vertical

vegetation belts has been added to the new subsection titled “Regional Setting, drilling, and core analyses.”

As highlighted in our response to comment #2 from the Reviewer, this elevation position indicates that the study site is highly suitable for investigating forest-meadow dynamics.

6. Figure 2b: The red and green curves are partially interlaced and reduce readability. It is recommended to change one of the colors.

We have updated the color scheme of the curves in Figure 2b by replacing the red color with purple for ETP curve and changing red vertical lines to orange. This change is intended to enhance readability.

7. To make it easier for the reader to understand and correspond to the text, please label the key vegetation transformation stages in Figure 3 and Figure 4.

We have highlighted the key vegetation transformation stages in Figure 3. While it is challenging to label these stages directly in Figure 4 due to the panel arrangement, we believe the inclusion of gray bars can assist readers in identifying and understanding these critical transitions.

8. Please ensure consistency in reference formatting. For instance, refs. 5, 12, and 18 appear to differ from the format of other references.

We have made the suggested revisions and thoroughly reviewed the reference list to ensure consistency in formatting for all references, adhering to the Formatting Requirements of *Nature Ecology & Evolution*.

Responses to the Reviews

We are grateful to both reviewers for their time in reviewing our manuscript. Their encouraging words are deeply appreciated.

Reviewer #2:

The authors have emphasized the reviewer's concerns, no more comments and suggestions.

Reviewer #3:

The authors have thoroughly addressed my previous concerns through expanded datasets, refined analyses, and clearer mechanistic interpretations. This work provides an exceptional perspective to reconstruct past vegetation dynamics with unprecedented and continuous 3.5-million-year pollen record, while offering actionable insights for modern vegetation conservation and future adaptation. I recommend publication in *Nature Ecology and Evolution* in its current form.